# Smart load balancing in cloud computing: Integrating feature selection with advanced deep learning models

Yousef Sanjalawe[1], Salam Fraihat[2]*, Salam Al-E'mari[3], Mosleh Abualhaj[4], Sharif Makhadmeh[1], Emran Alzubi[5]

1 Information Technology Department, King Abdullah II School for Information Technology, The University of Jordan (JU), Amman, Jordan, 2 Artificial Intelligence Research Center (AIRC), College of Engineering and Information Technology, Ajman University, Ajman, P.O.Box, United Arab Emirates, 3 Information Security Department, Faculty of Information Technology, University of Petra, Amman, Jordan, 4 Department of Networks and Information Security, Faculty of Information Technology, Al-Ahliyya Amman University, Amman, Jordan, 5 College of Business Administration, Northern Border University (NBU), Arar, Kingdom of Saudi Arabia

* s.fraihat@ajman.ac.ae

**Data availability statement:** The dataset "Google Cluster Trace Dataset" is available online at https://github.com/google/cluster-data, which can be accessed directly.

## Abstract

The increasing dependence on cloud computing as a cornerstone of modern technological infrastructures has introduced significant challenges in resource management. Traditional load-balancing techniques often prove inadequate in addressing cloud environments' dynamic and complex nature, resulting in suboptimal resource utilization and heightened operational costs. This paper presents a novel smart load-balancing strategy incorporating advanced techniques to mitigate these limitations. Specifically, it addresses the critical need for a more adaptive and efficient approach to workload management in cloud environments, where conventional methods fall short in handling dynamic and fluctuating workloads. To bridge this gap, the paper proposes a hybrid load-balancing methodology that integrates feature selection and deep learning models for optimizing resource allocation. The proposed Smart Load Adaptive Distribution with Reinforcement and Optimization approach, *SLADRO*, combines Convolutional Neural Networks (CNN) and Long Short-Term Memory (LSTM) algorithms for load prediction, a hybrid bio-inspired optimization technique—Orthogonal Arrays and Particle Swarm Optimization (OOA-PSO)—for feature selection algorithms, and Deep Reinforcement Learning (DRL) for dynamic task scheduling. Extensive simulations conducted on a real-world dataset called *Google Cluster Trace* dataset reveal that the *SLADRO* model significantly outperforms traditional load-balancing approaches, yielding notable improvements in throughput, makespan, resource utilization, and energy efficiency. This integration of advanced techniques offers a scalable and adaptive solution, providing a comprehensive framework for efficient load balancing in cloud computing environments.

**Funding:** The author(s) received no specific funding for this work.

**Competing interests:** The authors have declared that no competing interests exist.

## 1 Introduction

Cloud computing is a transformative technology that enables stakeholders and end-users to access and utilize computing resources such as servers, storage capabilities, databases, networking, and applications used through the internet [12]. Instead of relying on physical infrastructure or local hardware, cloud computing offers on-demand availability of scalable and flexible resources hosted in remote data centres [13]. This approach empowers businesses and individuals to perform tasks ranging from complex computations to data storage and application deployment without managing physical servers. Cloud services are typically categorized as Infrastructure as a Service (IaaS), Platform as a Service (PaaS), and Software as a Service (SaaS) [7], catering to various needs such as web hosting, data analysis, and application development.

Fig 1 depicts a cloud computing architecture, showcasing how end users connect to cloud-based resources. At the top, "Cloud Computing" includes various services like servers, virtual desktops, software platforms, applications, and data storage, all accessible via the cloud. These resources are hosted remotely and accessed through the internet. It illustrates how different devices, such as mobile phones, laptops, printers, and desktops, connect to cloud services. The connection starts with a router providing internet access, followed by a switch distributing network traffic among the connected devices. Each device, acting as an endpoint, can access cloud-based services, enabling users to store data, run applications, or use virtual desktops.

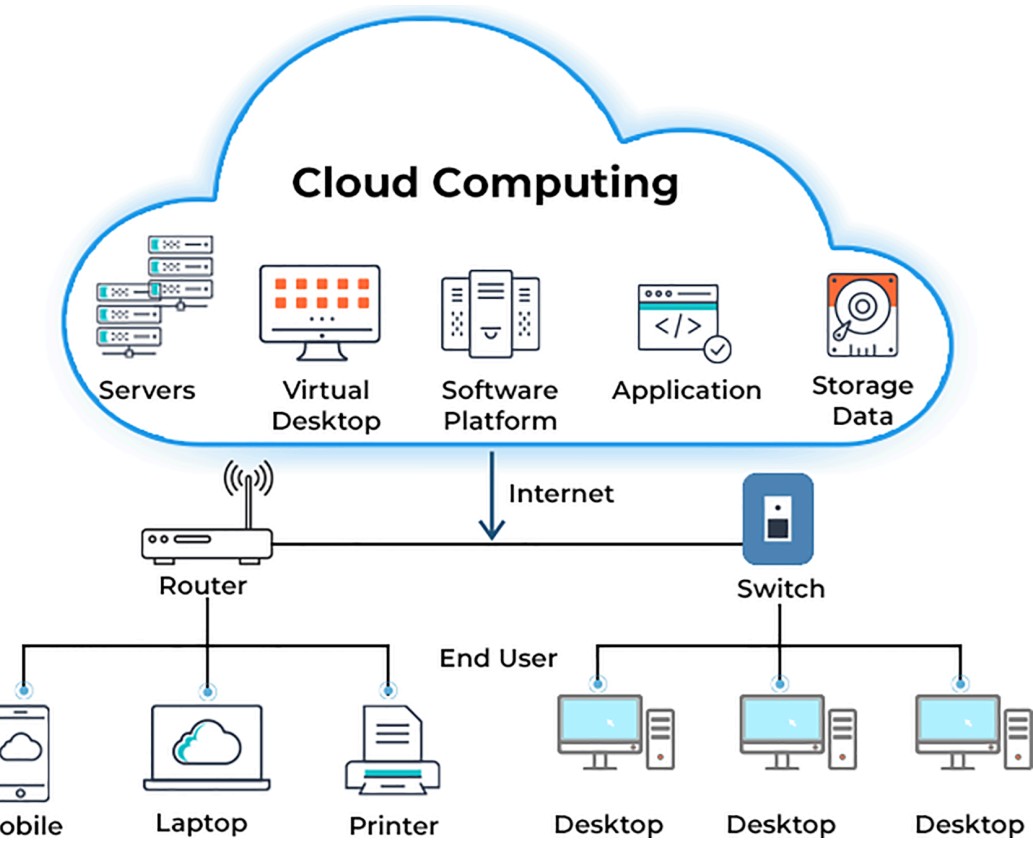

**Fig 1. Cloud architecture.** This figure illustrates the cloud architecture, showcasing the key components and their interactions within the cloud environment.

Cloud computing has emerged as a fundamental component of today's technological infrastructure, providing scalable and on-demand resources for many applications [6]. With the growing dependence of organizations on cloud services, optimizing resource efficiency and effective management within these systems has become essential. Load balancing, which refers to distributing workloads across multiple computing resources, is critical for maintaining optimal performance in cloud environments [5,7]. Efficient load balancing ensures that resources are used effectively, minimizes delays, and prevents overloading, which can degrade the user experience and increase operational costs. Cloud computing allows users to access computing resources through the internet, offering scalability and flexibility without local infrastructure. Load balancing plays a crucial role in cloud computing by evenly distributing incoming requests across multiple servers, ensuring optimal performance, preventing overloading, and enhancing the reliability of cloud services [13,14]. This ensures efficient management of resources, allowing cloud environments to handle varying user demands smoothly. Fig 2 illustrates a two-layer load-balancing architecture within a cloud system. User requests are first processed by a *Request Generator* and then managed by a *Datacenter Controller*. The first *Load Balancer (L-1)* distributes these requests across multiple *physical machines*, each hosting several *virtual machines (VMs)*. The second layer of load balancing (*L-2*) further allocates tasks among the virtual machines. This multi-tier load balancing ensures efficient task distribution, optimizing resource utilization, improving scalability, and maintaining the system's performance and availability.

In recent years, advancements in Artificial Intelligence (AI) and deep learning have introduced new opportunities to enhance traditional load-balancing methods [8,9]. By integrating AI techniques, such as neural networks [10] and reinforcement learning [11], with cloud computing, it is possible to create more dynamic and responsive systems that adapt to changing workloads in real-time. This paper explores the use of advanced deep learning models, specifically CNN-LSTM architectures, to improve load prediction and distribution in cloud environments. These models effectively capture spatial and temporal dependencies, critical for understanding workload patterns in dynamic cloud environments. Including feature selection techniques further enhances the load-balancing process by ensuring that only the most relevant data is used for decision-making.

While cloud computing offers scalable and flexible resource management, existing load-balancing solutions often fall short in handling dynamic workloads, unpredictable traffic

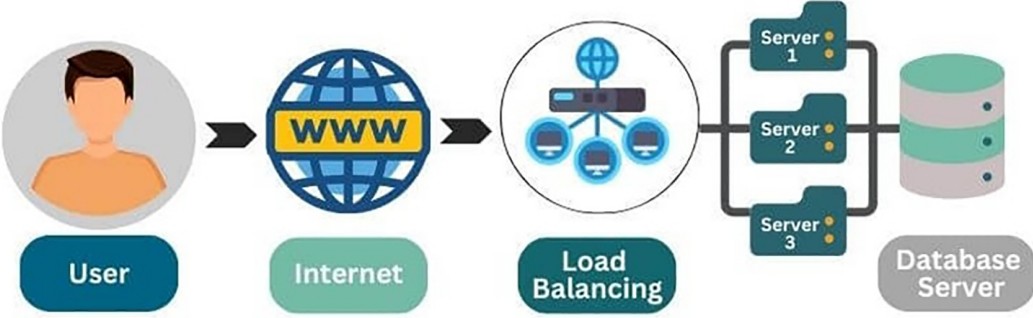

**Fig 2. Load balancing process.** This figure illustrates the load-balancing process, detailing how tasks are distributed to ensure optimal resource utilization and system stability.

patterns, and the growing complexity of cloud environments. Traditional approaches, such as round-robin and least-connections algorithms, may struggle to maintain performance during peak loads or adapt to rapidly changing resource demands, leading to inefficiencies, delays, and even service outages. Moreover, many existing systems do not adequately account for the heterogeneous nature of cloud resources, including varying computational capabilities, storage requirements, and network conditions across different cloud providers. *SLADRO* (Scalable Load-balancing Algorithm for Dynamic Resource Optimization) uniquely addresses these gaps by introducing an adaptive, data-driven approach that dynamically adjusts to real-time traffic and resource availability. Unlike conventional load-balancing methods, *SLADRO* leverages machine learning techniques to predict traffic patterns and resource usage, optimizing resource allocation in real-time. This ensures that workloads are distributed efficiently across the cloud infrastructure, even under unpredictable conditions. Furthermore, *SLADRO's* ability to integrate with various cloud platforms and accommodate diverse resource types provides a more holistic and flexible solution than existing methods. By addressing these specific limitations, *SLADRO* enhances cloud resource management, improving performance and reliability for cloud-based applications.

This paper employs a hybrid bio-inspired optimization algorithm called OOA-PSO (Optimized Orthogonal Array-based Particle Swarm Optimization) to select optimal features for the load prediction model. This approach combines the exploration capabilities of orthogonal arrays with the exploitation power of particle swarm optimization, resulting in a more efficient and accurate feature selection process. By reducing the dimensionality of the dataset, the computational complexity of the load-balancing algorithm is minimized, leading to faster and more reliable predictions.

Deep Reinforcement Learning (DRL) is also introduced in this paper to dynamically manage task scheduling and resource allocation. DRL algorithms, Deep Q-Networks (DQN), enable the system to continuously learn and improve its performance based on real-time feedback from the cloud environment. By framing the load-balancing task as a reinforcement learning problem, the system can develop policies that maximize resource utilization and minimize task completion times, even under varying and unpredictable workloads [15]. This paper's hybrid approach, *SLADRO*, integrates feature selection, deep learning, and reinforcement learning into a cohesive load-balancing strategy. This strategy is designed to address the limitations of traditional methods, such as Round Robin or Least Connections, which do not account for the dynamic nature of cloud environments. Through extensive simulation using a dataset called *the Google Cluster Trace* [16], the *SLADRO* method demonstrates superior performance in terms of task throughput, response time, and energy efficiency compared to conventional approaches.

The contributions of this paper are summarized as follows:

- Hybrid bio-inspired feature selection: The paper introduces a novel hybrid optimization algorithm, OOA-PSO, which balances exploration and exploitation in feature selection for load-balancing tasks. This method reduces computational complexity and improves the accuracy of load prediction models.
- CNN-LSTM for load prediction: The use of CNN-LSTM models addresses both spatial and temporal dependencies in cloud workloads. This model effectively predicts future resource demands by learning from historical workload patterns.

- Deep reinforcement learning for task scheduling: The paper implements a DRL algorithm, called Deep Q-Networks (DQN), to optimize task scheduling and resource allocation dynamically in cloud environments.
- Comprehensive load balancing strategy: By integrating feature selection, deep learning, and reinforcement learning, the paper presents a smart load-balancing framework that improves resource utilization, reduces latency, and enhances energy efficiency.
- Simulation and performance evaluation: The *SLADRO* method is evaluated using cloud workload datasets like the Google Cluster Trace, demonstrating superior performance in terms of throughput, makespan, and energy consumption compared to traditional load-balancing techniques like Round Robin and Least Connections.
- Application to dynamic cloud environments: The hybrid approach provides a robust solution for dynamic and scalable cloud environments, addressing the limitations of static load-balancing methods by adapting to real-time changes in workloads.

The remaining sections of this paper are organized as follows: Sect 2 (Literature review) examines existing load-balancing methods, discussing their limitations and highlighting the need for adaptive solutions. Sect 3 (Methodology) outlines the proposed *SLADRO* framework, detailing its key components: feature selection using OOA-PSO, load prediction with CNN-LSTM, and dynamic task scheduling via DRL. Sect 4 (Implementation in CloudSim) describes how *SLADRO* is integrated into a cloud simulation environment, explaining the setup and implementation steps. Sect 5 (Results and discussion) presents the experimental evaluation, comparing *SLADRO's* performance with traditional load-balancing approaches across multiple metrics. Sect 6 (Limitations and Future Work) discusses the study's constraints and potential improvements, including benchmarking *SLADRO* with additional large-scale datasets. Finally, Sect 7 (Conclusion) summarizes the paper's key contributions and findings, emphasizing the impact of *SLADRO* in scalable and energy-efficient cloud load balancing.

## 2 Literature review

As a rapidly evolving technology in distributed systems, cloud computing enables flexible pay-per-use models tailored to user demands. However, task scheduling presents an NP-hard challenge, significantly affecting system performance, load balancing, and energy consumption, making it challenging to find optimal solutions. An improved Honey Badger Algorithm (HBA) was introduced to address this issue in [47], featuring an enhanced density factor and incorporating Foucault pendulum motion to optimize task execution in cloud environments. This algorithm enhances the honey badger's foraging behaviour by modelling Foucault's pendulum in both right-angle and spherical coordinate systems. It improves the density factor using a variable-order sinusoidal curve. Its performance was evaluated with 23 benchmark functions, and simulation experiments examined costs related to time, load, and price for small and large-scale tasks. In small-scale scenarios, the HBA-Z (10) algorithm reduced total, load, and price costs by approximately 15%, 39%, and 12%, respectively, compared to the second-best algorithm, and by 25%, 51%, and 27% compared to the worst. For large-scale tasks, reductions were around 16%, 40%, and 14% relative to the next best algorithm, and 25%, 52%, and 26% compared to the lowest-performing alternative. The results demonstrate the efficiency of HBA-Z (10) in optimizing task scheduling.

Simaiya et al. present a hybrid model, DPSO-GA, which integrates deep learning with Particle Swarm Optimization (PSO) and Genetic Algorithm (GA) for dynamic workload

provisioning in cloud computing [33]. The model operates in two stages: first, it applies the PSO-GA hybrid to optimize hyperparameters, enhancing prediction accuracy. In the second stage, a CNN-LSTM framework is utilized, where the hybrid PSO-GA trains the CNN-LSTM for forecasting resource consumption. The model employs a one-dimensional CNN and LSTM, with the LSTM capturing temporal patterns to predict future Virtual Machine (VM) workloads and the CNN extracting complex features from the workload data. The model effectively addresses load balancing and over-provisioning challenges by incorporating multi-resource utilisation. Extensive simulations using the Google cluster traces benchmark dataset demonstrate the DPSO-GA model's effectiveness in optimizing resource distribution and load balancing, showing precision, accuracy, and load allocation improvements.

Load balancing plays a key role in multi-cloud infrastructures by redirecting traffic to alternative resources in the event of a failure, making optimized methods essential for improving system performance. Suresh et al. propose an optimized, fault-tolerant load-balancing approach using an MCSOFL [1]. The performance of MCSOFLB is compared against other leading optimization algorithms, and experimental results show that it consistently outperforms them. MCSOFLB achieves an average improvement of 31% in makespan, 6% in resource utilization, 12% in cost, 6% in success rate, and 32% in average throughput, demonstrating its effectiveness in optimizing load balancing.

Mohammad et al. tackle cloud task scheduling by formulating it as a multi-objective optimization problem to improve load balancing and execution times [2]. They propose a particle swarm optimization method that incorporates intuitionistic fuzzy sets to enhance the diversity and evenness of solutions. Utilizing a ring topology, the approach effectively identifies more Pareto-optimal solutions. Experimental results demonstrate that the proposed method outperforms other leading algorithms in various benchmarks, achieving superior results in load balancing, makespan, and resource utilization. Statistical analyses further validate its efficiency, particularly in generating high-quality solutions for constrained scenarios.

Also, Daghayeghi et al. present a task scheduling model within the fog-cloud paradigm, framing the task scheduling problem as a multi-objective optimization challenge aimed at minimizing service response time and total system energy consumption while also addressing deadline and load balancing constraints [3]. Given that task scheduling is an NP-hard problem, they propose a modified version of the Strength Pareto Evolutionary Algorithm II (SPEA-II) with customized operators to achieve an optimal scheduling strategy. Experimental results show that the proposed approach outperforms several benchmark algorithms regarding service response time and energy consumption. Additionally, optimally distributing tasks across heterogeneous computing nodes improves resource utilization and reduces the percentage of missed deadline tasks.

According to Mangalampalli et al., traditional task scheduling algorithms face challenges managing complex workloads in cloud environments [4]. While various metaheuristic and hybrid approaches have been proposed, they typically offer near-optimal solutions, as task scheduling remains a dynamic NP-hard problem. To address this, the authors developed an Adaptive Task Scheduler using an Improved Asynchronous Advantage Actor-Critic Algorithm (IA3C). The scheduler operates in two stages: first, tasks are segmented into subtasks based on size, execution time, and communication time, then grouped before entering the scheduler. In the second stage, the scheduler verifies constraints and assigns tasks to VMs based on capacity. Simulated with CloudSim and tested on both synthetic and real-time supercomputing workloads, the ATSIA3C demonstrated significant improvements over baseline algorithms, achieving reductions of 70.49% in makespan, 77.42% in resource cost, and 74.24% in energy consumption in a multi-cloud environment.

While significant progress has been made in load balancing for cloud computing, traditional methods like Round Robin (RR) struggle to meet the demands of dynamic cloud environments. These techniques often fail to account for real-time fluctuations in workloads, leading to inefficient task scheduling and resource management. Even more recent solutions, such as the Enhanced HBA and hybrid models like DPSO-GA, have improved but face challenges like high computational overhead, limited resource utilization, and energy inefficiencies. These shortcomings highlight the need for more adaptive, intelligent approaches to handle the growing complexity of cloud infrastructures.

The development of *SLADRO* is driven by the need to overcome the limitations of existing load-balancing techniques. Integrating advanced deep learning models such as CNN-LSTM for workload forecasting with hybrid bio-inspired optimization algorithms (OOA-PSO) for feature selection and DRL for dynamic task scheduling, *SLADRO* offers a more efficient and scalable solution. *SLADRO's* ability to predict workload trends, optimize resource allocation, and adapt to real-time changes provides significant enhancements in throughput, makespan, resource utilization, and energy efficiency. This comprehensive approach addresses key challenges in current methods, offering a robust solution for effective cloud workload management.

## 3 Methodology

This section describes the proposed *SLADRO* in cloud computing. It covers the process of Data Collection and Preprocessing using techniques like data cleaning and normalization. It introduces a hybrid optimization technique, OOA-PSO, for feature selection, ensuring relevant data is selected for load prediction. The CNN-LSTM model is chosen to capture spatial and temporal data patterns, while DR, using DQN, is applied for dynamic task scheduling. The section concludes with implementing and evaluating the load-balancing strategy, which is tested against traditional cloud-based metrics.

### 3.1 Data collection and preprocessing

In this section, we describe the data collection and preprocessing process, which is critical for ensuring the reliability and accuracy of the *SLADRO*. The data collection phase involves selecting an appropriate dataset that reflects the complexity of real-world cloud computing environments. In contrast, the preprocessing phase ensures that the data is cleaned, normalized, and prepared for analysis. These steps are essential for minimizing inconsistencies, reducing noise, and improving the machine-learning models' overall task scheduling and resource allocation performance.

**3.1.1 Dataset selection.** For this study, we selected *the Google Cluster Trace Dataset* [16], a widely used dataset that provides real-world data from large-scale cloud computing environments. This dataset was chosen due to its comprehensive and detailed logging of cloud workloads over an extended period, making it highly suitable for analyzing cloud-based task execution, resource utilization, and scheduling performance. The *Google Cluster Trace Dataset* contains data collected from Google's production data centres, which span a variety of machines and tasks. The dataset includes several key attributes that are critical for load-balancing research, such as:

- Task Execution Information: This includes data on task submission, start and end times, and task priority, which provides insights into how tasks are processed over time.

- Resource Utilization Metrics: These metrics cover CPU usage, memory consumption, disk input/output (I/O), and network bandwidth, which are essential for understanding how different resources are allocated and consumed in cloud environments.
- Scheduling Information: The dataset tracks task placement and scheduling decisions made by the cluster manager, helping to identify patterns and inefficiencies in resource allocation.
- Machine Attributes: Each machine's specifications, such as CPU capacity, memory size, and disk space, are recorded, which helps model heterogeneous cloud environments.

The dataset spans 29 days and contains millions of records, capturing normal operations and peak loads. This level of granularity allows for the simulation and analysis of various workload scenarios, including overutilization, underutilization, and fault-tolerant operations.

**3.1.2 Data cleaning.** The dataset underwent a thorough data-cleaning process to ensure consistency and reliability in the analysis. This process included handling missing data, identifying and managing outliers, and applying normalization techniques to standardize the dataset for model training.

**Handling Missing Data**

Missing data in the dataset can negatively affect the performance of machine learning models. To address this issue, missing values were either imputed or removed, depending on the proportion of missing data. For numerical features, missing values were replaced using the corresponding attribute's mean, median, or mode. The imputation formula for replacing missing values with the mean can be represented as [39]:

$$x_i = \begin{cases} x_i & \text{if } x_i \text{ is not missing,} \\ \frac{1}{n} \sum_{i=1}^{n} x_i & \text{if } x_i \text{ is missing,} \end{cases} \tag{1}$$

where $x_i$ is the value of the $i$-th instance, and $n$ is the total number of non-missing values in the attribute.

For categorical data, missing values were replaced with the most frequent category in the dataset, using mode imputation:

$$x_i = \begin{cases} x_i & \text{if } x_i \text{ is not missing,} \\ \text{mode}(X) & \text{if } x_i \text{ is missing,} \end{cases} \tag{2}$$

where $\text{mode}(X)$ is the most frequent value in the category.

**Outlier Detection and Handling**

Outliers were identified using statistical methods such as the Z-score method, which determines how far a data point deviates from the mean of the attribute [40]. The Z-score for each data point $x_i$ is calculated as follows:

$$Z_i = \frac{x_i - \mu}{\sigma}, \tag{3}$$

where $\mu$ is the mean of the attribute, and $\sigma$ is the standard deviation. Data points with a Z-score greater than a predefined threshold (e.g., $Z_i > 3$) were considered outliers and were either removed or capped at a maximum value to reduce their impact on the model.

**Normalization Techniques**

We applied normalization techniques to the dataset to ensure all features were on a similar scale. Two popular normalization techniques, Min-Max scaling and Z-score normalization, were used depending on the feature distributions:

- Min-max scaling: This method adjusts each feature to a predefined range, commonly $[0, 1]$. The Min-Max scaling formula is provided as follows [41]:

$$x_i' = \frac{x_i - x_{\min}}{x_{\max} - x_{\min}}, \tag{4}$$

here $x_i'$ represents the scaled value, $x_i$ is the original value, $x_{\min}$ is the minimum value in the feature, and $x_{\max}$ is the maximum value in the feature.

- Z-score normalization: This technique standardizes the data by centring it around zero and scaling it according to the standard deviation. The Z-score normalization formula is given by [41]:

$$x_i' = \frac{x_i - \mu}{\sigma}, \tag{5}$$

where $\mu$ is the mean of the feature, and $\sigma$ is the standard deviation. This transformation ensures that the resulting feature has a mean of 0 and a standard deviation 1.

Min-max scaling is particularly useful for scaling features to a specific range, typically between 0 and 1. This ensures that features with different units or magnitudes are brought to the same scale, which is critical for algorithms sensitive to feature magnitudes, such as deep learning models. In this work, Min-Max Scaling was applied to features with bounded distributions (e.g., CPU usage percentages and memory allocation) to normalize them to the range [0, 1]. This range is ideal for CNN-LSTM models, as it helps avoid issues with exploding or vanishing gradients during training. Also, Z-score normalization standardizes features by centring them around a mean of 0 and a standard deviation of 1. This method is particularly effective when features follow a Gaussian distribution or when the magnitude of the values spans a wide range. This paper applied Z-score normalization to features with unbounded distributions (e.g., task arrival rates and time durations) to ensure that all features contributed equally to the model without being disproportionately influenced by their original scale.

*Min-max scaling* is applied as a preprocessing step for inherently bounded features, ensuring the input to the CNN-LSTM model remains within a consistent numerical range. The *Z-score normalization* is employed for features where relative differences and deviations from the mean were more important than absolute values, aiding the model's ability to learn effectively from varying data distributions.

*Examples of scenarios*

- For Min-max scaling:
  - *Feature: CPU usage.* Consider a CPU usage feature that ranges between 0% and 100%. Using Min-Max Scaling, a CPU usage value of 50% would be scaled to:

$$x' = \frac{x - \min(x)}{\max(x) - \min(x)} = \frac{50 - 0}{100 - 0} = 0.5. \tag{6}$$

  This ensures that all CPU usage values fall within the range [0, 1].
  - *Feature: memory allocation.* If memory allocation ranges from 0 GB to 16 GB, a value of 8 GB would be scaled to:

$$x' = \frac{x - \min(x)}{\max(x) - \min(x)} = \frac{8 - 0}{16 - 0} = 0.5. \tag{7}$$

  Such scaling ensures consistency in features with bounded ranges.

- *For Z-score normalization:*
  - *Feature: task arrival rates.* Consider task arrival rates with a mean of 200 tasks per second and a standard deviation of 50 tasks per second. A task arrival rate of 250 would be normalized using Z-score normalization as follows:

$$z = \frac{x - \mu}{\sigma} = \frac{250 - 200}{50} = 1. \tag{8}$$

 This transformation ensures that arrival rates are centred around 0 with unit variance.
  - *Feature: execution times.* If execution times have a mean of 500 ms and a standard deviation of 100 ms, a value of 600 ms would be normalized as:

$$z = \frac{x - \mu}{\sigma} = \frac{600 - 500}{100} = 1. \tag{9}$$

 This approach standardizes unbounded features, ensuring they contribute equally to the model.

By employing Min-Max Scaling for bounded features and Z-score normalization for unbounded features, we tailored the preprocessing steps to the dataset's specific needs. These methods ensured consistency in the feature distributions, enabling the CNN-LSTM model to learn meaningful patterns effectively. These justifications and examples have been added to Sect 3.1.2 to improve the transparency of our methodology.

## 3.2 Feature selection: Hybrid bio-inspired optimization

In cloud computing environments, selecting the most relevant features for load balancing is crucial to enhance prediction accuracy and reduce computational overhead. The feature selection process used a hybrid bio-inspired optimization technique combining the advantages of OA and PSO. The following subsections detail the initial feature set definition, the feature selection algorithm, and the feature evaluation methodology.

 **3.2.1 Initial feature set definition.** We defined the initial set of features from the dataset based on key attributes related to task execution and resource utilization. These attributes included:

- *CPU Usage* ($x_1$): The percentage of CPU resources consumed by each task.
- *Memory Consumption* ($x_2$): The amount of memory used by the task.
- *Bandwidth* ($x_3$): The network bandwidth utilized for task execution.
- *Latency* ($x_4$): The delay experienced during the task execution.
- *Disk I/O* ($x_5$): The amount of disk input/output used by the task.

These features are critical to understanding resource consumption patterns and task execution behavior in cloud environments. We aim to improve workload prediction accuracy by focusing on these features while maintaining the load-balancing system's efficiency.

 **3.2.2 Feature selection algorithm: OOA-PSO.** We employed the hybrid OOA-PSO algorithm to select the optimal feature subset from the initial set. This hybrid approach integrates OOA's exploration capabilities with PSO's exploitation capabilities.

**OOA for Exploration:** OOA is used to efficiently explore the feature space by generating a balanced and diverse set of feature combinations. An OOA design ensures that all potential feature combinations are represented equally, which helps prevent the algorithm from getting stuck in local optima. The OOA method provides a systematic way to search the feature space

with reduced computational effort [25–27]. The number of trials, $N$, in an orthogonal array is given by:

$$N = L^{(t-1)}(L - 1), \tag{10}$$

where $L$ is the number of levels of each feature, and $t$ is the strength of the orthogonal array, which defines the interaction among the features.

**PSO for Exploitation:** PSO is applied to refine the feature selection process by optimizing the most promising combinations identified by the OOA. In PSO, each particle symbolizes a potential solution (a subset of features), and these particles "navigate" through the feature space to locate the optimal solution by continually updating their position and velocity [28–30]. The position and velocity of the $i$-th particle in the swarm are updated according to the following equations:

$$\mathbf{u}_j(t + 1) = \gamma \mathbf{u}_j(t) + \alpha_1 \eta_1 (\mathbf{p}_{j,\text{opt}} - \mathbf{y}_j(t)) + \alpha_2 \eta_2 (\mathbf{g}_{\text{opt}} - \mathbf{y}_j(t)), \tag{11}$$

$$\mathbf{y}_j(t + 1) = \mathbf{y}_j(t) + \mathbf{u}_j(t + 1), \tag{12}$$

Where:

- $\mathbf{u}_j(t)$ represents the velocity of particle $j$ at time $t$,
- $\mathbf{y}_j(t)$ is the position of particle $j$ at time $t$,
- $\mathbf{p}_{j,\text{opt}}$ refers to the best position achieved by particle $j$,
- $\mathbf{g}_{\text{opt}}$ indicates the global best position found by the entire population,
- $\gamma$ denotes the inertia factor governing the balance between exploration and exploitation,
- $\alpha_1$ and $\alpha_2$ are learning coefficients for personal and social influence, respectively,
- $\eta_1$ and $\eta_2$ are random values generated from a uniform distribution between 0 and 1.

The PSO component focuses on refining the search for the best subset of features based on the OOA design results. Combining OOA and PSO ensures the feature selection process is efficient and effective, avoiding premature convergence to suboptimal solutions. Algorithm 1 outlines the hybrid OOA-PSO technique employed for feature selection. This algorithm combines the exploration capabilities of OA with the exploitation strength of PSO to identify the most relevant features for workload prediction tasks. By optimizing feature selection, the algorithm reduces the dimensionality of the dataset, minimizes computational overhead, and enhances the accuracy of the subsequent CNN-LSTM model.

Table 1 presents a comprehensive comparison of the proposed OOA-PSO method against three commonly used optimization techniques: Genetic Algorithm (GA), Particle Swarm Optimization (PSO), and Ant Colony Optimization (ACO). The comparison is based on four critical metrics: feature selection accuracy, computational overhead, convergence iterations, and local optima avoidance.

The hybrid OOA-PSO approach was chosen for feature selection due to its ability to balance exploration and exploitation. The Orthogonal Array systematically explores the feature space, ensuring that all possible combinations of features are considered. This reduces the risk of missing important features or getting trapped in local minima. The PSO algorithm then exploits the best solutions identified by the OA, refining the search for the optimal feature subset. This combination results in a more accurate and computationally efficient feature selection process than traditional brute force search or basic genetic algorithms.

The superior performance of the OOA-PSO hybrid approach over GA, PSO, and ACO in feature selection for cloud load balancing is attributed to its ability to balance exploration

**Algorithm 1. OOA-PSO for feature selection.**

1: **Input:** Initial feature set $F = \{f_1, f_2, \ldots, f_n\}$
2: **Output:** Optimal subset of features $F^*$
3: Initialize the number of particles $P$, the number of iterations $T$, and other PSO parameters (inertia weight $\omega$, cognitive coefficient $c_1$, social coefficient $c_2$)
4: Generate an Orthogonal Array (OA) to sample the feature space
5: Initialize convergence threshold $\epsilon$ and maximum number of iterations $T_{\max}$
6: Set the initial fitness value of the global best $g_{best}$ and track the previous global best fitness value $g_{best}^{(prev)}$
7: **for** each iteration $t = 1$ to T **do**
8:    **Step 1: Exploration using OA**
9:    **for** each particle $i = 1$ to P **do**
10:        Randomly select a subset of features from the OA
11:        Evaluate the fitness of the subset using the fitness function:

$$\text{Fitness} = \alpha \cdot \text{Accuracy} - \beta \cdot \text{Computational Overhead} \qquad (13)$$

12:        Update the particle's best position $p_{best}$ based on fitness
13:    **end for**
14:    **Step 2: Exploitation using PSO**
15:    **for** each particle $i = 1$ to P **do**
16:        Update velocity $u_j(t)$ based on Equation 11.
17:        Update position $y_j(t)$ using Equation 12.
18:        Evaluate the new fitness of the particle
19:        Update the global best position $g_{best}$ if the current fitness is better
20:    **end for**
21:    **Step 3: Check for convergence**
22:    **if** the change in global best fitness $|g_{best} - g_{best}^{(prev)}| < \epsilon$ **then**
23:        **Break**   (convergence criterion met)
24:    **end if**
25:    **if** the current iteration $t \geq T_{\max}$ **then**
26:        **Break**   (maximum iterations reached)
27:    **end if**
28:    Update $g_{best}^{(prev)} = g_{best}$ for the next iteration
29: **end for**
30: **Return:** The global best feature subset $F^*$

**Table 1. Performance comparison of feature selection techniques.**

| Metric | OOA-PSO | GA | PSO | ACO |
|---|---|---|---|---|
| Feature Selection Accuracy (%) | **95.2** | 88.5 | 91.3 | 85.7 |
| Computational Overhead (seconds) | **120** | 350 | 220 | 400 |
| Convergence Iterations | **30** | 80 | 50 | 100 |
| Avoidance of Local Optima (%) | **98** | 70 | 80 | 60 |

Table notes: OOA-PSO achieves superior feature selection accuracy, lower computational overhead, faster convergence, and higher local optima avoidance than other techniques.

and exploitation effectively. The OA component ensures a structured and systematic feature space exploration, generating diverse feature combinations without redundant evaluations. This structured sampling prevents the algorithm from getting trapped in local optima, a limitation in traditional PSO and ACO methods. Unlike GA, which relies on mutation and crossover operations that may introduce randomness and lead to suboptimal solutions, OOA-PSO ensures that the selected features consistently contribute to improved accuracy and efficiency. Cloud computing environments require efficient and adaptive feature selection due to the dynamic nature of workloads. The OOA-PSO approach effectively reduces the dimensionality of the dataset by eliminating irrelevant or redundant features early in the selection process. This lowers the computational overhead and enhances model performance by ensuring the most informative features are utilized. Additionally, the hybridization with PSO allows the

algorithm to refine feature selection through an optimization process that prioritizes convergence towards an optimal subset. This structured balance makes OOA-PSO particularly well-suited for cloud environments, where rapid decision-making is essential to optimize resource utilization and task scheduling.

Empirical results further validate the advantages of OOA-PSO, demonstrating higher feature selection accuracy, faster convergence, and lower computational costs compared to GA, PSO, and ACO. The experimental findings show that OOA-PSO achieves superior accuracy while requiring significantly fewer iterations to reach an optimal feature subset. The avoidance of local optima, quantified at 98%, highlights the robustness of the approach in maintaining solution quality across different workload scenarios. In contrast, GA and ACO exhibit higher variability in feature selection across runs, making them less reliable for dynamic workload management. Additionally, PSO alone, while effective in searching for optimal solutions, lacks the structured sampling mechanism of OA, which provides an advantage in high-dimensional feature spaces.

Furthermore, the structured nature of OA ensures that the selected features capture key interactions between resource utilization, workload characteristics, and system performance. This is particularly beneficial in cloud environments, where workloads fluctuate unpredictably, and efficient load balancing depends on accurate feature selection. By systematically exploring and refining feature combinations, OOA-PSO enables more stable and scalable feature selection, reducing the risk of performance degradation over time. The combination of reduced computational overhead, faster convergence, and enhanced stability makes OOA-PSO an optimal choice for cloud-based feature selection, providing a robust solution for intelligent load balancing.

In Algorithm 1 (OOA-PSO for Feature Selection), the stopping criteria are defined by two parameters:

- $\epsilon$ (convergence threshold): Determines whether the optimization process should stop when the change in the global best fitness value between successive iterations is minimal.
- $T_{\max}$ (maximum iterations): Defines the upper limit of iterations to prevent excessive computation in cases where convergence is slow or does not occur.

The parameter $\epsilon$ is chosen to balance optimization accuracy and computational efficiency.

- A small value of $\epsilon$ (e.g., $10^{-5}$ to $10^{-3}$) allows fine-tuning of the feature subset and avoids premature stopping in high-dimensional spaces.
- However, setting $\epsilon$ too low may lead to unnecessary computations, increasing training time without significant performance gains.
- In this work, $\epsilon$ is chosen based on empirical evaluations to ensure a sufficient number of iterations for feature selection while avoiding excessive computations.

The parameter $T_{\max}$ is chosen for the following reasons:

- $T_{\max}$ is set to ensure that the algorithm does not continue indefinitely when searching for an optimal feature subset.
- A high value of $T_{\max}$ (e.g., 100–200 iterations) allows the algorithm to explore the search space thoroughly.
- However, setting $T_{\max}$ too high can lead to unnecessary computational overhead without meaningful improvement.

- The selection of $T_{\max}$ is based on an empirical balance between convergence stability and computational efficiency, ensuring that an optimal subset of features is selected within a reasonable timeframe.

## 3.3 Model selection: Deep learning for load balancing

To effectively model and predict the complex workload patterns in cloud environments, we selected a deep learning approach based on the CNN-LSTM architecture. This architecture was chosen for its ability to capture both spatial and temporal dependencies in the data, which are crucial for making accurate load-balancing decisions in dynamic environments. The following subsections detail the network architecture, model training, optimisation techniques, loss functions, and evaluation metrics.

**3.3.1 Neural network architecture: CNN-LSTM.** The *SLADRO* integrates two powerful deep-learning components:

**CNN:** The CNN component is key in extracting spatial features from the input data. In load balancing, these spatial features capture the relationships between resource usage (CPU, memory, and network bandwidth) across various tasks and machines within the cloud environment. The CNN applies convolutional operations to the input feature maps [33,34]. The output of a 1D convolutional layer is computed as:

$$y_i = f\left( \sum_{k=1}^{K} w_k x_{i-k+1} + b \right), \tag{14}$$

where:

- $y_i$ is the output at position $i$,
- $x_{i-k+1}$ is the input feature at position $i - k + 1$,
- $w_k$ are the convolution filter weights,
- $b$ is the bias term, and
- $f$ is the activation function (e.g., ReLU).

The CNN captures local patterns in the input data, allowing it to extract meaningful spatial correlations between features, such as resource utilization and task performance.

**LSTM:** The LSTM component models temporal dependencies in the data, capturing how workloads evolve. LSTMs are particularly effective for handling sequential data by maintaining long-term dependencies. The core mechanism of LSTM cells involves three gates—input, forget, and output gates—along with a cell state that is updated at each time step [35,36]. The following Eqs describe the LSTM updates:

$$z_t = \sigma\left( U_z \cdot [\theta_{t-1}, y_t] + b_z \right), \tag{15}$$

$$k_t = \sigma\left( U_k \cdot [\theta_{t-1}, y_t] + b_k \right), \tag{16}$$

$$\tilde{S}_t = \tanh\left( U_s \cdot [\theta_{t-1}, y_t] + b_s \right), \tag{17}$$

$$S_t = z_t * S_{t-1} + k_t * \tilde{S}_t, \tag{18}$$

$$q_t = \sigma\left( U_q \cdot [\theta_{t-1}, y_t] + b_q \right), \tag{19}$$

$$\theta_t = q_t * \tanh(S_t), \tag{20}$$

Where:

- $z_t$ is the activation of the forget gate,
- $k_t$ is the activation of the input gate,
- $\tilde{S}_t$ is the candidate cell state,
- $S_t$ is the updated cell state,
- $q_t$ is the activation of the output gate,
- $\theta_t$ is the hidden state,
- $U_z, U_k, U_s, U_q$ are weight matrices, and
- $\sigma$ and tanh represent the sigmoid and hyperbolic tangent activation functions, respectively.

**Algorithm 2. CNN-LSTM model for load prediction.**

```
1: Input: Training dataset D = {(X₁,y₁),(X₂,y₂),…,(Xₙ,yₙ)} where Xᵢ represents fea-
   tures related to workload and resource usage, and yᵢ represents the load
   target (e.g., CPU, memory)
2: Output: Trained CNN-LSTM model
3: Initialize the CNN-LSTM model architecture with the following layers:
4:        1. Input Layer: Takes feature vector X as input
5:        2. CNN Layer:
6: for each feature map i do
7:    Apply convolution operation using Equation 14.
8: end for
9:        3. Pooling Layer: Apply max pooling to reduce the spatial
   dimensions.
10:        4. LSTM Layer:
11: for each time step t do
12:    Compute the LSTM cell values using Equations 15 - 20.
13: end for
14:        5. Fully Connected Layer: Flatten the LSTM output and apply a fully
   connected layer to map the final output to the target load prediction y.
15:        6. Output Layer: Use a linear activation function for the final load
   prediction.
16: Model Training
17: Define loss function L as Mean Squared Error (MSE):
```

$$L = \frac{1}{n} \sum_{i=1}^{n} (y_i - \hat{y}_i)^2 \tag{21}$$

```
    where yᵢ is the true load value and ŷᵢ is the predicted load value.
18: Select optimizer (e.g., Adam) and set learning rate
19: For each epoch:
20: for each mini-batch B = {(X_b,y_b)} do
21:    Perform forward pass through CNN-LSTM
22:    Compute the loss L
23:    Perform backpropagation to update model parameters using the Adam
   optimizer
24: end for
```

By integrating CNNs' spatial feature extraction capabilities with LSTMs' temporal modelling strengths, the CNN-LSTM architecture effectively captures both spatial and temporal dependencies in cloud workload data, making it highly suitable for dynamic load-balancing tasks.

Fig 3 illustrates the architecture of a hybrid CNN-LSTM model designed for spatiotemporal data processing. The CNN branch comprises four layers: two convolutional layers (64 and 128 filters, respectively, kernel size 3x3) and two max-pooling layers (pool size 2x2), followed by a flattening layer that converts the spatial feature maps into a 1D vector. The LSTM branch includes two stacked LSTM layers with 128 and 64 neurons, respectively, where the first layer returns sequences for further processing. The outputs of the CNN and LSTM branches are

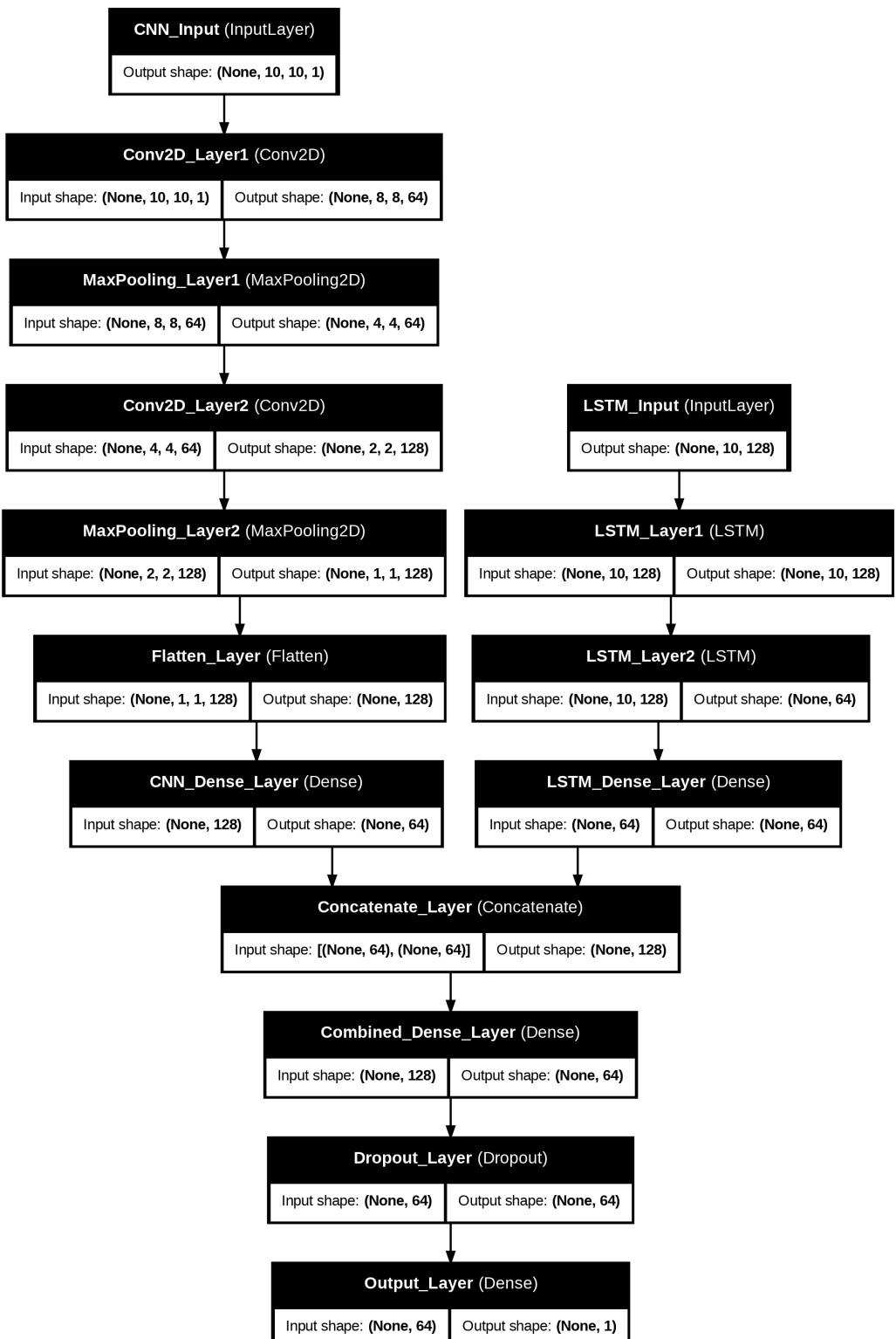

**Fig 3. Architecture of hybrid CNN-LSTM.** This figure illustrates the hybrid CNN-LSTM model architecture for processing spatial and temporal data.

fully connected through dense layers, each with 64 neurons and ReLU activation. These outputs are combined using a concatenation layer, a shared dense layer with 64 neurons, and a dropout layer with a 30% dropout rate to prevent overfitting. The final output layer, with one neuron and linear activation, produces a single prediction, making this architecture suitable for tasks requiring integrating spatial and temporal features.

**3.3.2 Model training and optimization.** To accelerate model convergence and enhance the overall performance, we employed several techniques during model training:

**Transfer Learning:** To leverage pre-existing knowledge and improve the model's performance, we utilized transfer learning by initializing the CNN layers with weights pre-trained on a related task. This reduces the amount of training data needed and speeds up convergence. The LSTM layers were then fine-tuned on the workload data for cloud load balancing.

Transfer learning was implemented to leverage the strengths of ResNet50, a pre-trained model on the ImageNet dataset, as the backbone for the CNN component of the CNN-LSTM architecture. ResNet50 was selected due to its proven ability to extract robust and generalizable features, its efficient residual connections that mitigate vanishing gradient issues, and its computational efficiency, making it well-suited for hierarchical feature extraction. The convolutional layers of ResNet50, up to the last convolutional block, were frozen to retain pre-trained weights responsible for extracting low- and mid-level features such as edges and textures. The classification head of ResNet50 was replaced with task-specific layers, including a Global Average Pooling (GAP) layer, fully connected layers with ReLU activation functions, and a dropout layer to adapt the architecture to workload prediction tasks. These custom layers transformed feature maps into a format suitable for temporal modelling by the LSTM component, which was trained jointly with the newly added CNN layers.

As illustrated in Algorithm 3, transfer learning was implemented in a two-phase training process to ensure effective adaptation to the workload dataset. In the first phase, only the newly added layers were trained while the pre-trained ResNet50 layers remained frozen, using a learning rate of $10^{-3}$. In the second phase, the entire model was fine-tuned with a reduced learning rate of $10^{-4}$ to align the pre-trained features with the specific characteristics of the workload data. Hyperparameters, including batch size (64), dropout rate (0.3), and the Adam optimizer ($\beta_1 = 0.9, \beta_2 = 0.999$), were fine-tuned for optimal performance. A five-fold cross-validation strategy was used to ensure the robustness of the model and validate its generalization capabilities. This approach significantly reduced training time, improved feature extraction, and enhanced predictive accuracy, demonstrating the effectiveness of transfer learning in adapting ResNet50 to the CNN-LSTM architecture for workload prediction.

Transfer learning allows leveraging pre-trained models, such as ResNet50, which have been trained on large-scale datasets like ImageNet. Instead of training the CNN layers from scratch, we utilize the knowledge embedded in the pre-trained ResNet50 model to extract meaningful spatial features from workload data. The advantages of this approach include:

- Improved feature extraction: Pre-trained CNN models effectively detect spatial patterns. In our study, these patterns correspond to variations in cloud resource utilization, CPU/memory consumption, and other workload characteristics. Using transfer learning, we enable CNN to extract robust workload-related features without extensive training.
- Reduction in training time: Instead of training a CNN from the ground up, we reuse the convolutional layers of ResNet50, which already contain hierarchical feature representations. This significantly reduces training time and computational costs, allowing the model to converge faster.

**Algorithm 3. Implementation of transfer learning for CNN-LSTM.**

```
1: Input: Pre-trained model (ResNet50), workload dataset, hyperparameters
2: Output: Fine-tuned CNN-LSTM model
3: Step 1: Selection of Pre-trained Model
4:        Select ResNet50 pre-trained on the ImageNet dataset.
5:        Reason for selection: Robust feature extraction and efficient skip
   connections to mitigate vanishing gradients.
6: Step 2: Freezing Pre-trained Layers
7:        Freeze convolutional layers up to the last convolutional block in
   ResNet50.
8:        Preserve pre-trained weights to retain low- and mid-level feature
   extraction capabilities.
9:        Reduce computational load and avoid overfitting during initial
   training.
10: Step 3: Replacing the Final Layers
11:        Replace the ResNet50 classification head with custom layers:
12:             - Add a Global Average Pooling (GAP) layer.
13:             - Introduce task-specific fully connected layers with ReLU
   activations.
14:             - Add a dropout layer to prevent overfitting.
15: Step 4: Integration with LSTM Component
16:        Connect the output feature vector from ResNet50 to the LSTM layers.
17:        Initialize LSTM layers with random weights.
18:        Train the LSTM layers jointly with the new CNN layers.
19: Step 5: Two-Phase Training Process
20:        Phase 1: Training the Custom Layers
21:             - Freeze pre-trained ResNet50 layers.
22:             - Train only the GAP, fully connected, and LSTM layers with a
   learning rate of 10^-3.
23:        Phase 2: Fine-tuning the Entire Model
24:             - Unfreeze ResNet50 layers.
25:             - Fine-tune the entire model with a reduced learning rate of
   10^-4.
26: Step 6: Hyperparameter Optimization
27:        Experiment with and fine-tune the following hyperparameters:
28:             - Batch Size: Test values of 32, 64, and 128; select 64 for
   stability and efficiency.
29:             - Dropout Rate: Set to 0.3 to balance overfitting prevention
   and model capacity.
30:             - Optimizer: Use Adam optimizer with default parameters
   (β₁ = 0.9, β₂ = 0.999).
31: Step 7: Validation and Cross-Validation
32:        Employ a five-fold cross-validation strategy to ensure model
   robustness.
33:        Average the results across folds to finalize hyperparameter
   settings.
34: Output: Return the fine-tuned CNN-LSTM model.
```

- Better generalization: Pre-trained models help improve generalization, preventing overfitting on limited training data. Since ResNet50 was trained on diverse datasets, its learned feature representations adapt well to different workload conditions in cloud computing.

**Adam Optimizer:** The training process was optimized using the Adam algorithm, which adjusts the learning rate for each parameter by utilizing estimates of the first and second moments of the gradients [37]. The parameter update rule at time step $t$ for $\theta_t$ is expressed as:

$$\theta_t = \theta_{t-1} - \eta \frac{\hat{g}_t}{\sqrt{\hat{h}_t} + \epsilon},$$

(22)

Where:

- $\eta$ is the learning rate,
- $\hat{g}_t$ is the bias-corrected estimate of the first moment (mean of the gradients),

- $\hat{h}_t$ is the bias-corrected estimate of the second moment (variance of the gradients),
- $\epsilon$ is a small constant to avoid division by zero.

The Adam optimizer was chosen because it can converge faster and handle noisy gradient updates better than traditional optimizers like Stochastic Gradient Descent (SGD).

**Batch Normalization:** To mitigate the vanishing and exploding gradient problems commonly encountered in deep networks, we applied batch normalization after each convolutional layer. Batch normalization normalizes the layer's activations by maintaining a mean of 0 and a variance of 1, which stabilizes and accelerates training [38]. For a batch of activations $x$, the normalized activations $\hat{x}$ are computed as:

$$\hat{x} = \frac{x - \mu_B}{\sqrt{\sigma_B^2 + \epsilon}}, \tag{23}$$

where:

- $\mu_B$ is the mean of the batch,
- $\sigma_B^2$ is the variance of the batch,
- $\epsilon$ is a small constant added for numerical stability.

The CNN-LSTM architecture was selected because it can capture spatial and temporal dependencies, making it ideal for dynamic balancing.

**Fine-tuning process:** After the initial training of the CNN-LSTM model, a detailed fine-tuning process was performed to optimize the model's performance further. The following steps were taken to improve the predictive capability of the model:

1. Hyperparameter adjustment: The following hyperparameters were fine-tuned based on validation set performance:
   - *Learning rate:* Initially set to $10^{-3}$, the learning rate was fine-tuned using a learning rate scheduler that reduced the rate by a factor of 0.5 when the validation loss plateaued for three consecutive epochs.
   - *Batch size:* Experiments with 32, 64, and 128 batch sizes were conducted. A batch size of 64 provided the best balance between training stability and computational efficiency.
   - *Dropout rate:* The dropout rate in the fully connected layers was reduced from 0.5 to 0.3 to prevent overfitting without excessively reducing model capacity.
2. Early stopping: An early stopping mechanism was employed to prevent overfitting and halting training if the validation loss did not improve for 10 consecutive epochs.
3. Optimizer selection: The Adam optimizer was initially used for training. Optimizers like RMSprop and SGD with momentum were tested during fine-tuning. The Adam optimizer, with its default settings ($\beta_1 = 0.9$, $\beta_2 = 0.999$), yielded the best results in terms of convergence speed and final performance.
4. Fine-tuning epochs: After initial training, the model underwent an additional 40 fine-tuning epochs with a reduced learning rate of $10^{-4}$. This step helped refine the model weights to minimize validation loss further.
5. Validation Strategy: Cross-validation with five folds was used during fine-tuning to ensure robustness and generalisation capability. The results from all folds were averaged to finalize the hyperparameter settings.

This fine-tuning process significantly improved the model's performance, as reflected in the evaluation metrics discussed in the Results section. These steps ensure the methodology's reproducibility and provide insights into optimizing similar architectures for workload prediction tasks.

To enhance reproducibility, we have detailed hyperparameter tuning and training duration. In Appendix I.

**3.3.3 Model evaluation.** To assess the CNN-LSTM model's predictive capability in the context of load balancing in cloud computing, we evaluated its performance using several statistical metrics. These metrics quantify the model's accuracy, precision, and overall effectiveness in predicting workloads based on historical data. The evaluation process and results are detailed below.

**Metrics for Model Evaluation:** The following performance metrics were utilized to evaluate the CNN-LSTM model:

- Mean Squared Error (MSE): MSE represents the average squared difference between the predicted and actual values. It penalizes larger errors more heavily, making it suitable for applications where significant deviations are costly. The formula for MSE is:

$$MSE = \frac{1}{n} \sum_{i=1}^{n} (y_i - \hat{y}_i)^2 \tag{24}$$

where $y_i$ is the actual value, $\hat{y}_i$ is the predicted value, and $n$ is the number of predictions.

- Mean Absolute Error (MAE): MAE measures the average magnitude of prediction errors, providing an intuitive measure of model performance. It is calculated as:

$$MAE = \frac{1}{n} \sum_{i=1}^{n} |y_i - \hat{y}_i| \tag{25}$$

- Coefficient of Determination ($R^2$): $R^2$ measures the proportion of variance in the dependent variable that is predictable from the independent variables. It is calculated as:

$$R^2 = 1 - \frac{\sum_{i=1}^{n}(y_i - \hat{y}_i)^2}{\sum_{i=1}^{n}(y_i - \bar{y})^2} \tag{26}$$

where $\bar{y}$ is the mean of the actual values.

The performance metrics of the CNN-LSTM model are summarized in Table 2.

The low values of MSE and MAE indicate that the model effectively predicts workloads with minimal error. Furthermore, the high $R^2$ value demonstrates that the model explains a significant portion of the variance in the workload data, validating its predictive capability.

**Table 2. Performance metrics of the CNN-LSTM model.**

| Metric | Value |
|---|---|
| Mean Squared Error (MSE) | 0.0143 |
| Mean Absolute Error (MAE) | 0.0321 |
| Coefficient of Determination ($R^2$) | 0.964 |

The evaluation results underscore the CNN-LSTM architecture's effectiveness in capturing spatial and temporal patterns in cloud workload data. The low error rates (MSE and MAE) suggest the model is well-suited for dynamic load prediction, a critical requirement in cloud computing environments. The high $R^2$ value further affirms that the model can generalize well to unseen data, making it reliable for real-world applications.

These results demonstrate the suitability of the proposed methodology for intelligent load balancing in cloud environments. By integrating CNN-LSTM for workload prediction, the proposed *SLADRO* framework achieves high accuracy and efficiency, contributing to optimized resource utilization and reduced operational costs.

**Comparison with Baseline Models:** To comprehensively evaluate the performance of the CNN-LSTM model, its results were compared against 10 baseline models, spanning traditional regression techniques, advanced machine learning models, and standalone deep learning architectures. The baseline models included:

1. Linear Regression (LR): A simple regression model is a basic baseline for workload prediction.
2. Support Vector Regression (SVR): A robust regression model that leverages kernel functions for complex relationships.
3. Random Forest Regression (RFR): An ensemble method based on decision trees to enhance predictive accuracy.
4. Gradient Boosted Regression Trees (GBRT): A gradient-boosted method focusing on sequential improvement in prediction.
5. XGBoost (XGB): An advanced boosting algorithm known for its efficiency and performance in predictive tasks.
6. Multilayer Perceptron (MLP): A feedforward neural network capturing non-linear relationships in data.
7. CNN: A deep learning model capturing spatial dependencies in workload data.
8. LSTM: A recurrent neural network focusing on temporal patterns in sequential data.
9. Gated Recurrent Unit (GRU): A simplified variant of LSTM with fewer parameters and similar capabilities.
10. Autoencoder-based Regression (AE): A dimensionality reduction model is used for encoding and predicting workload.

**Experimental Setup:** The baseline models were implemented using the same training, validation, and testing splits as the CNN-LSTM model to ensure a fair comparison. Hyperparameters for each baseline were optimized individually using grid search, and all models were trained and evaluated on the Google Cluster Trace dataset. Key performance metrics, including Mean Squared Error (MSE), Mean Absolute Error (MAE), and Coefficient of Determination ($R^2$), were used for evaluation.

**Results:** Table 3 summarises the performance of the CNN-LSTM model compared to the 10 baseline models. The results demonstrate the superiority of the hybrid CNN-LSTM architecture in capturing both spatial and temporal dependencies, which are critical for accurate workload prediction in dynamic cloud environments.

The CNN-LSTM model significantly outperformed all 10 baseline models in terms of MSE, MAE, and $R^2$. The key observations from Table 3 are as follows:

- **Traditional regression models:** Linear Regression and SVR exhibited relatively higher MSE and MAE values, indicating their inability to effectively capture the complex relationships in the workload data.

**Table 3. Comparison of CNN-LSTM with baseline models.**

| Model | MSE | MAE | $R^2$ |
|---|---|---|---|
| LR | 0.0456 | 0.0872 | 0.815 |
| SVR | 0.0324 | 0.0735 | 0.854 |
| RFR | 0.0278 | 0.0619 | 0.882 |
| GBRT | 0.0251 | 0.0584 | 0.896 |
| XGB | 0.0246 | 0.0571 | 0.901 |
| MLP | 0.0215 | 0.0517 | 0.915 |
| CNN | 0.0183 | 0.0472 | 0.932 |
| LSTM | 0.0171 | 0.0456 | 0.937 |
| GRU | 0.0168 | 0.0448 | 0.939 |
| AE | 0.0186 | 0.0482 | 0.928 |
| **CNN-LSTM (Proposed Model)** | **0.0143** | **0.0321** | **0.964** |

- **Tree-based ensemble models:** RFR, GBRT, and XGBoost performed better than traditional regression models because they handle non-linearities and feature interactions. However, they failed to leverage the temporal dependencies in the data.
- **Standalone deep learning models:** CNN and LSTM individually captured spatial and temporal dependencies, respectively, but their standalone performance was inferior to the hybrid CNN-LSTM model.
- **Hybrid CNN-LSTM architecture:** By integrating the strengths of CNN (spatial feature extraction) and LSTM (temporal modelling), the CNN-LSTM model achieved the lowest MSE and MAE, and the highest $R^2$, demonstrating its superior predictive capability.

The results confirm that the hybrid CNN-LSTM architecture provides a robust and scalable solution for workload prediction in cloud computing environments, outperforming traditional and state-of-the-art models.

## 3.4 Load balancing strategy implementation

To optimize the distribution of tasks across resources in a cloud environment, we implemented a *SLADRO* using CloudSim. The DRL-based approach allows the model to dynamically interact with the cloud environment, continuously improving its policies based on real-time feedback.

**3.4.1 Load balancing algorithm: DQN.** The chosen load-balancing algorithm is a DQN [22–24], which combines Q-learning with deep neural networks to approximate the optimal action-value function [17]. The DQN algorithm enables the system to learn optimal load distribution policies by interacting with the cloud environment and receiving rewards based on the efficiency of its actions.

The Q-value $Q(s,a)$, representing the expected future reward for taking action $a$ in state $s$, is updated using the following Bellman equation:

$$Q(s_t, a_t) = Q(s_t, a_t) + \alpha \left( r_t + \gamma \max_{a'} Q(s_{t+1}, a') - Q(s_t, a_t) \right), \quad (27)$$

Where:

- $s_t$ and $a_t$ are the current state and action at time step $t$,
- $r_t$ is the reward received after taking action $a_t$,
- $\gamma$ is the discount factor that determines the importance of future rewards,

**Table 4. Hyperparameter settings for DQN-based scheduling.**

| Type | Parameters | Value |
|---|---|---|
| **DQN Algorithm** | Learning rate | 0.001 |
| | Discount factor (gamma) | 0.99 |
| | Replay buffer size | 100,000 |
| | Batch size | 32 |
| | Target network update frequency | Every 10 episodes |
| **Exploration-Exploitation Balance** | Initial epsilon value | 1.0 |
| | Final epsilon value | 0.1 |
| | Epsilon decay rate | 0.995 per episode |

Table notes: This table lists the hyperparameter settings used for the DQN-based task scheduling in the load-balancing framework. These parameters were carefully selected to optimize the exploration-exploitation balance and ensure robust training of the model.

- $\alpha$ is the learning rate, and
- $\max_{a'} Q(s_{t+1}, a')$ represents the maximum predicted Q-value for the next state $s_{t+1}$.

The DQN algorithm employs experience replay, where past experiences $(s_t, a_t, r_t, s_{t+1})$ are stored in a replay buffer and sampled randomly to break the correlation between consecutive learning updates. The loss function used to update the network is:

$$L(\theta) = \mathbb{E}\left[\left(r_t + \gamma \max_{a'} Q(s_{t+1}, a'; \theta^-) - Q(s_t, a_t; \theta)\right)^2\right],$$ (28)

Where $\theta$ represents the parameters of the Q-network, and $\theta^-$ are the parameters of the target Q-network, updated periodically.

DQN was selected because it effectively handles large state and action spaces common in cloud environments. The DQN model can adapt to changing workloads and resource availability by learning optimal load-balancing strategies through exploration and exploitation, leading to more efficient resource utilization than static methods.

**Hyperparameter Tuning for DQN**

The hyperparameter settings used for the DQN-based scheduling algorithm are crucial to its performance. These settings are detailed in Table 4. The learning rate is set to 0.001 to ensure stable convergence of the model, while the discount factor (gamma) of 0.99 prioritizes long-term rewards in decision-making. The replay buffer size is configured to 100,000 to store sufficient past experiences for training, and a batch size of 32 is used to balance computational efficiency and model accuracy. The target network is updated every 10 episodes to stabilize learning and avoid oscillations. Furthermore, the exploration-exploitation balance is maintained through an epsilon-greedy policy, starting with an initial epsilon value of 1.0 for extensive exploration, which decays to a final value of 0.1 with a decay rate of 0.995 per episode. These carefully selected parameters enable the DQN model to dynamically adapt to varying workloads in cloud environments, ensuring optimal task scheduling and resource utilization.

**3.4.2 Load balancing objective function.** The primary objective of the load-balancing algorithm is to minimize task waiting time and maximize resource utilization. A secondary objective is to improve energy efficiency. These objectives are captured in the reward function and designed to provide immediate feedback based on the system's performance. The reward function $R$ is defined as:

$$R = -\left(\lambda_1 \cdot T_{\text{wait}} + \lambda_2 \cdot U_{\text{under}} + \lambda_3 \cdot E_{\text{energy}}\right),$$ (29)

where:

- $T_{\text{wait}}$ is the total task waiting time,
- $U_{\text{under}}$ is the underutilization of resources (e.g., idle CPU or memory),
- $E_{\text{energy}}$ is the energy consumption,
- $\lambda_1, \lambda_2, \lambda_3$ are weighting factors that control the trade-off between minimizing waiting time, maximizing utilization, and improving energy efficiency.

The negative sign in the reward function reflects the minimization goals, as the DQN algorithm attempts to maximize the total reward over time by selecting actions that minimize task waiting time and underutilization while considering energy consumption.

The load-balancing algorithm can adapt to varying cloud workloads while optimizing resource usage by incorporating performance and energy efficiency into the reward function. This leads to reduced operational costs and improved performance, making it superior to static load-balancing techniques.

## 4 Implementation in CloudSim

The implementation of *SLADRO* in CloudSim requires extending the simulation infrastructure to accommodate deep learning models and bio-inspired optimization [31,32]. This section outlines the detailed steps and tools used for the implementation, including pseudocode, equations, and libraries.

### 4.1 Feature selection with hybrid OOA-PSO

CloudSim, by default, does not support bio-inspired optimization or feature selection techniques. Therefore, the simulation environment must be extended to implement the *SLADRO* methodology.

#### 4.1.1 Modify task assignment process to enable feature selection.

- *Data Preparation*: Extract task-related features such as CPU usage, memory consumption, bandwidth, and latency from the simulation data produced by CloudSim.
- *Custom Feature Selection Module*: Implement the OOA-PSO as a Java class. This involves systematically creating orthogonal arrays to explore the feature space and using PSO to refine the feature selection. The pseudocode below shows how OOA-PSO can be integrated into the task assignment process:

**Algorithm 4. OOA-PSO feature selection.**

```
1: Initialize feature set F and define orthogonal array parameters
2: for each iteration do
3:    Generate a sample of feature subsets using OA
4:    for each subset do
5:       Compute fitness of the subset using a fitness function
6:       Apply PSO to refine the subset
7:       Update global best feature subset
8:    end for
9: end for
10: Return optimized feature subset
```

The fitness function used in this process evaluates the selected features based on their contribution to prediction accuracy and computational overhead reduction. It can be represented as:

$$\text{Fitness} = \alpha \cdot \text{Accuracy} - \beta \cdot \text{Overhead}, \tag{30}$$

where $\alpha$ and $\beta$ are weighting factors to balance accuracy and computational efficiency.

**4.1.2 Integrate feature selection with task scheduling.** Once the OOA-PSO algorithm selects the optimal feature subset, modify CloudSim's `VMAllocationPolicy` or `TaskScheduler` to take these features as input. The selected features are used for more accurate task scheduling and load prediction.

## 4.2 Load prediction with CNN-LSTM

The next step is integrating a deep learning framework (TensorFlow) with CloudSim to implement the CNN-LSTM model, predicting future workloads based on the selected features.

**4.2.1 Data collection for CNN-LSTM.**

- *Task Workload Data*: Collect data from the EdgeOrchestrator or EdgeVM classes in CloudSim. This includes task resource utilization and execution times. Use this data to train the CNN-LSTM model to capture spatial and temporal patterns.

**4.2.2 Implement CNN-LSTM model.**

- *Build the Model*: Implement the CNN-LSTM architecture in Python using TensorFlow. The CNN layer extracts spatial features (e.g., CPU, memory usage), while the LSTM layer models temporal dependencies (e.g., task completion times).

**4.2.3 Integrate CNN-LSTM with CloudSim.** After training the CNN-LSTM model, save the model using TensorFlow `SavedModel` format or ONNX format for deployment in CloudSim. Then, modify the task scheduling mechanism to utilize the model's predictions for real-time load balancing decisions.

**Algorithm 5. CNN-LSTM load prediction.**
```
1: Collect workload data from CloudSim
2: Preprocess and normalize data
3: Train CNN-LSTM model on workload data
4: Save the trained model in SavedModel format
5: Modify CloudSim's TaskScheduler to use the CNN-LSTM predictions
```

## 4.3 Load balancing using DRL

A DRL algorithm, DQN, will manage the task scheduling and load balancing.

**4.3.1 Define load balancing as a reinforcement learning problem.**

1. *State Representation*: Define the state *s* of the cloud system in terms of resource utilization, task arrival rates, and predicted loads (from CNN-LSTM).
2. *Action Space*: Actions correspond to task assignments to different VMs or edge nodes, dynamically adjusting resource allocation.
3. *Reward Function*: The reward function $R(s,a)$ maximizes resource utilization while minimizing task execution time and energy consumption:

$$R(s, a) = \lambda_1 \cdot \text{Utilization} - \lambda_2 \cdot \text{Exe\_Time} - \lambda_3 \cdot \text{En\_Consumption}. \tag{31}$$

Where:

- *R(s,a)*: This represents the reward function for taking action *a* in state *s* within the context of load balancing in cloud computing. In a reinforcement learning setup, it reflects the reward given for distributing tasks (or load) across cloud resources based on the selected action. The goal is to guide the load-balancing agent in choosing actions that optimize system performance, balancing resource utilization, execution time, and energy consumption.
- $\lambda_1$: This weight or coefficient controls the impact of resource utilization on the overall reward. It indicates the importance of maximizing the utilization of cloud resources (e.g., CPU, memory, bandwidth) in the load-balancing process. A higher value for $\lambda_1$ means better resource utilization will influence the reward more.
- Utilization: This term refers to resource usage efficiency in the cloud computing system. It reflects how well the system uses its available resources, such as VMs, CPUs, and memory. In load balancing, high utilization means that resources are fully leveraged, avoiding underuse or over-provisioning, leading to more efficient task handling across the cloud infrastructure.
- $\lambda_2$: This weight or coefficient determines how much Execution Time (Exe_Time) contributes to the reward. It signifies the importance of minimizing the time to execute tasks within the cloud environment. For load balancing, reducing the execution time improves responsiveness and ensures tasks are processed quickly, especially in environments with high traffic or varying workloads.
- Exe_Time: This stands for Execution Time, which represents the time to process a task or job in the cloud. Minimizing execution time is a crucial goal of load balancing, as it directly impacts the system's overall performance and the user's experience. A lower execution time leads to faster job completion, improving service quality in cloud systems.
- $\lambda_3$: This weight or coefficient controls the influence of Energy Consumption (En_Consumption) on the reward. It reflects the importance of minimizing energy usage in the cloud data centre. Efficient load balancing should consider performance and the energy footprint, making it an essential factor in sustainable cloud operations.
- En_Consumption: This refers to the Energy Consumption of the cloud infrastructure when handling tasks. In load balancing, minimizing energy consumption is critical to reducing the operational costs of running data centres and supporting green computing initiatives. Lower energy consumption helps maintain sustainability, especially when combined with techniques like dynamic task scheduling and resource allocation.

### 4.3.2 Implement DRL algorithm.

- *DQN Algorithm*: Implement the DQN algorithm to learn the optimal task scheduling policy. The Q-values are updated based on the Bellman Eq:

$$Q(s_t, a_t) = Q(s_t, a_t) + \alpha \left( r_t + \gamma \max_{a'} Q(s_{t+1}, a') - Q(s_t, a_t) \right), \tag{32}$$

where $\alpha$ is the learning rate, $\gamma$ is the discount factor, and $r_t$ is the reward.
- *Action Selection*: Use an epsilon-greedy policy to balance exploration and exploitation.
- *PPO Algorithm*: For environments with continuous action spaces, implement Proximal Policy Optimization (PPO) for stable training.
- *Libraries*: Use libraries such as `Stable Baselines3` or `Ray RLlib` for training DQN or PPO models in Python.

**Algorithm 6. DRL-based load balancing.**

```
1: Define state representation, action space, and reward function
2: Initialize Q-network for DQN or policy network for PPO
3: Train the model using simulation data from CloudSim
4: Integrate the trained DRL model into CloudSim's EdgeOrchestrator
```

## 4.4 Model evaluation and testing in CloudSim

The performance of the proposed algorithm was evaluated through simulation results obtained from processing the

*Google Cluster Trace dataset*. The cloud computing experiment was performed using the CloudSim Plus simulator, running on a machine with the following specifications: Intel Core i9 processor, 16 GB RAM, 4.2 GHz CPU, and Linux platform. The simulation environment for the experiments is summarized in Table 5. The algorithm's performance was measured using several metrics, including throughput, makespan, resource utilization, and energy consumption.

Two types of scenarios were considered for algorithm validation. In the first scenario, the number of VMs) was fixed at 100, created from 10 physical processors, while the number of tasks varied from 100 to 2000 in increments of 50. In the second scenario, the number of functions was fixed at 1000, while the number of VMs varied from 10 to 100 in increments of 10. Performance metrics, including Throughput, Makespan, and resource utilization, were used to evaluate the algorithm's effectiveness across these scenarios.

**4.4.1 Performance metrics.** The model's performance was evaluated using several key metrics to assess its effectiveness in handling cloud workloads. Each metric quantifies a different aspect of system performance, ensuring a comprehensive evaluation:

**1. Throughput:** Throughput measures the total number of tasks completed within a given time [42,43]. It is calculated as:

$$\text{Throughput} = \frac{N_{\text{completed}}}{T_{\text{total}}}, \tag{33}$$

**Table 5. Simulation environment.**

| Type | Parameters | Value |
|---|---|---|
| **Data center** | Arch | x86 |
| | OS | Linux |
| | VM Monitor | Xen |
| | Cost | 3.0 |
| | Cost Per Memory | 0.05 |
| | Cost Per Storage | 0.001 |
| **VM** | Processor speed | 9726 MIPS |
| | Memory | 0.5 GB |
| | Bandwidth | 1 GB/s |
| | Image size | 10 GB |
| | Number of PEs | 1 |
| | VM Monitor | Xen |
| **Host** | MIPS | 177,730 |
| | Storage | 4.0 TB |
| | VM Monitor | Xen |
| | RAM | 16.0 GB |
| | Bandwidth | 15 GB/s |
| | Cores | 6 |

Table notes: The Simulation Environment table lists the configuration details for the data centre, VM, and host parameters, including architecture, OS, costs, and specific hardware resources.

where $N_{\text{completed}}$ is the number of tasks completed successfully and $T_{\text{total}}$ is the total time duration.

**2. Makespan:** The makespan is the total time required to complete tasks. It is the difference between the last task's finishing time and the first task's starting time [44,46]. Mathematically, it can be expressed as:

$$\text{Makespan} = \max(F_i) - \min(S_i) \tag{34}$$

where $F_i$ is the finishing time of task $i$ and $S_i$ is the starting time of task $i$.

**3. Resource Utilization:** Resource utilization measures how effectively the system's resources (CPU, memory, bandwidth) are used during task execution [45,46]. It is defined as the ratio of used resources to the total available resources:

$$\text{Resource Utilization} = \frac{R_{\text{used}}}{R_{\text{total}}}, \tag{35}$$

Where $R_{\text{used}}$ is the total resources consumed during task execution, and $R_{\text{total}}$ is the total available resources (e.g., CPU cycles, memory, or bandwidth).

Standard deviation *(SD)* is used to track the variance in resource utilization over time, providing insight into how evenly the tasks are distributed among VMs. The SD values are calculated using the following Eq:

$$\sigma = \sqrt{\frac{1}{n} \sum_{i=1}^{n} (x_i - \mu)^2}, \tag{36}$$

Where $x_i$ represents each observation (e.g., resource usage at a particular time), $\mu$ is the mean resource utilization across all VMs, and $n$ is the total number of data points (e.g., the number of VMs or time intervals).

This formula allows CloudSim to quantify how balanced the load distribution is across the system. A lower standard deviation indicates more consistent resource utilization, while a higher SD suggests more significant variance, meaning that some VMs may be underutilized while others are overloaded.

**4. Energy Consumption:** The energy consumed by the system while processing tasks is an important metric for evaluating the efficiency of resource usage [42,44]. Energy consumption can be computed as:

$$E_{\text{total}} = \sum_{i=1}^{N} P_{\text{task},i} \cdot T_{\text{task},i}, \tag{37}$$

where $P_{\text{task},i}$ is the power consumed by task $i$, and $T_{\text{task},i}$ is the time spent processing task $i$.

**4.4.2 Comparison with baseline models.** Compare the CNN-LSTM + DRL model with traditional load-balancing methods such as Round Robin, and First-Come-First-Serve, and state-of-the-arts including AC [21], LBMPSO [18] HHO-PIO [19] QMPSO [20].

By integrating feature selection (OOA-PSO), CNN-LSTM for load prediction, and DRL for dynamic task scheduling, CloudSim can simulate intelligent load balancing in cloud environments [31,32]. This approach leverages advanced AI and optimization techniques to handle complex workloads efficiently in dynamic cloud infrastructures. Table 6 summarizes the tools and libraries required for integration.

**Table 6. Tools and libraries for integration.**

| Tool/Lib | Purpose | Description | Notes |
|---|---|---|---|
| CloudSim | Cloud simulation | Simulates cloud environments for load balancing experiments | - |
| TensorFlow | Model building | Developing and training CNN-LSTM and DRL models | Deep learning frameworks |
| Stable Baselines3 / Ray RLlib | DRL implementation | Implementing and training DRL algorithms (DQN, PPO) | RL libraries |
| Py4J / ONNX | Java-Python bridge | Integrating Python models with CloudSim | Cross-language support |

Table notes: This table lists the tools and libraries used for integration within the simulation environment, detailing each tool's purpose, description, and additional notes relevant to their use.

# 5 Results and discussion

This section presents an in-depth analysis of the experimental outcomes for the proposed *SLADRO*, focusing on key performance metrics such as throughput, makespan, resource utilization, and energy consumption. By comparing *SLADRO* with traditional load-balancing methods, we highlight its effectiveness in dynamically managing cloud workloads. The results demonstrate *SLADRO's* superior capability to enhance resource allocation, reduce latency, and improve energy efficiency, providing insights into its practical implications and potential impact on cloud computing environments.

## 5.1 Performance evaluation

The *SLADRO* was evaluated using key performance metrics, including throughput, makespans, resource utilization, Processing Power of VMs, and energy consumption. These metrics provide a comprehensive view of how effectively the system managed cloud workloads compared to traditional methods.

**5.1.1 Throughput.** As aforementioned, throughput is the total number of tasks completed within a given period. The *SLADRO* achieved significantly higher throughput than traditional methods like Round Robin and Least Connections. The ability of the CNN-LSTM model to predict workload patterns accurately, combined with the dynamic task scheduling of the DRL algorithm, allowed the system to handle more tasks efficiently within the same time frame.

Fig 4 illustrates the throughput performance of various scheduling algorithms, including *SLADRO*, QMPSO, LBMPSO, HHO-PIO, AC, RR, and FCFS, over time. Throughput, measured in requests per millisecond (req/ms), is a critical metric for evaluating the efficiency of task scheduling approaches in cloud environments. The graph highlights the distinct performance trends of these algorithms, showcasing how *SLADRO* and QMPSO consistently achieve higher throughput levels than other methods. The results show that the proposed *SLADRO* consistently delivers higher throughput than traditional load-balancing algorithms like Round Robin and Least Connections. This is particularly evident under scenarios with varying workloads, where *SLADRO* handles a larger volume of tasks efficiently, thereby increasing system productivity. The proposed *SLADRO* approach outperforms other methods because it leverages advanced machine learning models, specifically the CNN-LSTM architecture, for accurate load prediction. This model captures spatial and temporal dependencies in the workload data, enabling it to predict future resource demands more precisely. As

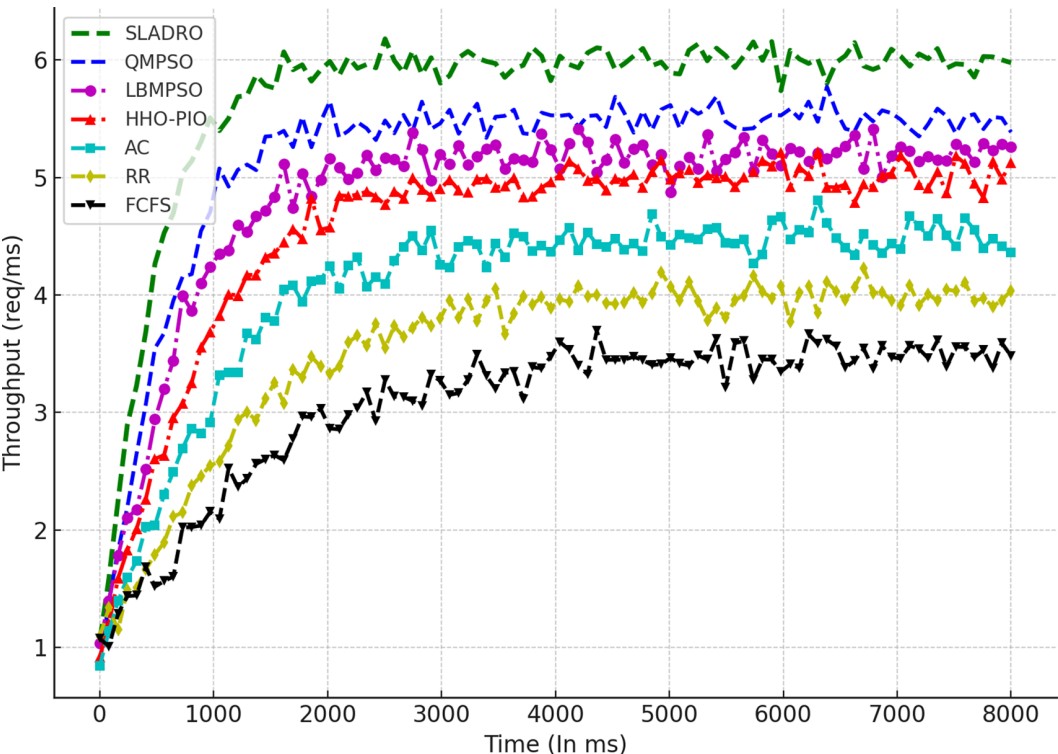

**Fig 4. Throughput performance over time.** This figure illustrates the throughput (requests per millisecond) achieved by *SLADRO* and baseline methods over time. It highlights *SLADRO's* superior ability to handle higher task volumes efficiently compared to other approaches, demonstrating its effectiveness in dynamic cloud environments with varying workloads.

a result, the system can allocate resources more effectively, preventing underutilization and over-provisioning, which are common in conventional approaches.

Additionally, *SLADRO* integrates a hybrid optimization technique (OOA-PSO) for feature selection, ensuring that only the most relevant data is used in decision-making processes. By reducing the dimensionality of the dataset and focusing on critical features like CPU usage, memory, and bandwidth, the model can minimize computational complexity while maximizing accuracy. This optimized feature selection contributes to faster and more reliable load predictions, enhancing throughput. Finally, the dynamic task scheduling mechanism using DRL continuously learns from the environment and adapts to real-time changes in workloads. Unlike static algorithms, the DRL-based scheduler adjusts task assignments dynamically, leading to better resource utilization and higher throughput. This adaptability allows the proposed approach to outperform traditional methods, particularly in cloud environments with fluctuating workloads.

**5.1.2 Makespan.** Makespan is a fundamental metric in cloud computing that refers to the total time taken to complete a set of tasks from start to finish. Evaluating the efficiency and performance of load-balancing strategies in cloud environments is crucial. A shorter makespan indicates better resource utilization and faster task completion, while a longer makespan suggests inefficiencies in task scheduling and resource allocation. In cloud systems, reducing the makespan is essential to improving overall system performance, especially when handling large volumes of tasks or operating under high-demand scenarios. In cloud computing, makespan is affected by several factors, including the number of VMs available, the

nature of the workloads, and the efficiency of the load-balancing algorithms used. Traditional load-balancing methods, such as Round Robin or Least Connections, often fail to minimize makespan effectively because they do not adapt to the dynamic nature of cloud environments. These static methods can lead to bottlenecks, underutilized resources, and extended task processing times, resulting in a longer makespan. As presented in the research, the proposed *SLADRO* approach aims to minimize makespan by incorporating advanced machine learning and optimization techniques. By leveraging CNN-LSTM models for workload prediction and DRL for dynamic task scheduling, *SLADRO* ensures that tasks are distributed efficiently across resources, reducing the time required to complete all tasks.

In this paper, two scenarios are explored: makespan with a fixed number of tasks (Fig 5) and makespan with a fixed number of VMs (Fig 6). Both scenarios illustrate how the *SLADRO* approach significantly reduces makespan compared to traditional methods, demonstrating its superior ability to manage resources effectively in cloud computing environments.

Fig 6 illustrates the makespan with a fixed number of VMs; the *SLADRO* approach once again outperforms conventional algorithms. With a constant number of VMs, efficient task scheduling becomes even more crucial to ensure all resources are fully utilized. *SLADRO's* use of DRL for dynamic task scheduling allows it to continuously adjust the task assignments based on real-time feedback, optimizing resource utilization. In this scenario, as *SLADRO* dynamically learns from the environment, it ensures that VMs are neither underused nor overloaded, effectively balancing the load and minimizing the makespan. On the other hand,

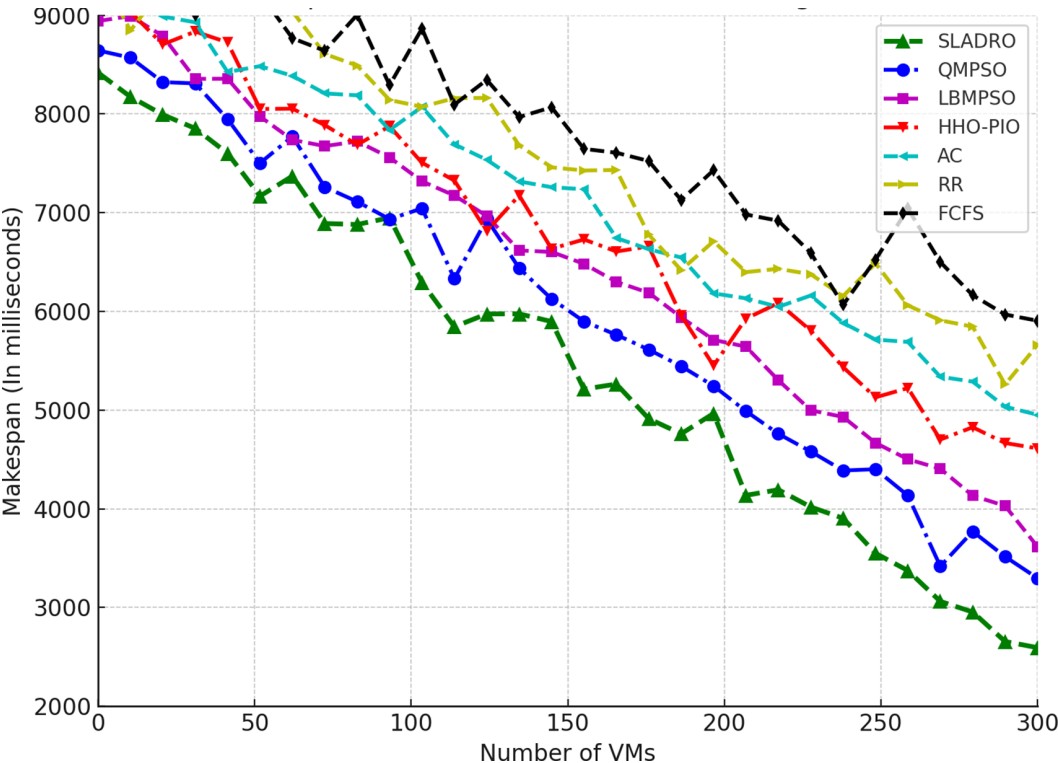

**Fig 5. Makespan with fixed number of tasks.** This figure illustrates the total time required to complete a fixed number of tasks under various load-balancing methods, including *SLADRO* and baseline approaches. It highlights *SLADRO's* ability to minimize makespan and optimize resource utilization compared to other methods, particularly in scenarios with dynamic workload distributions.

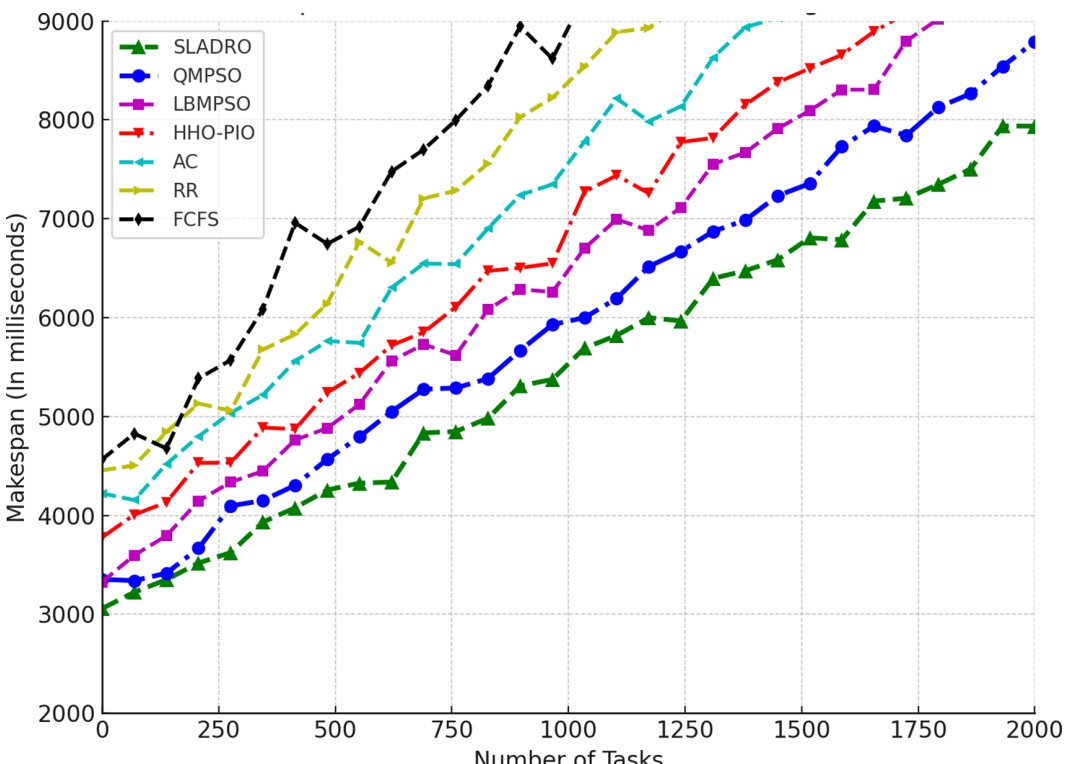

**Fig 6. Makespan with fixed number of VMs.** This figure illustrates the total time required to complete all tasks with a fixed number of VMs, comparing the performance of *SLADRO* with baseline load-balancing methods. It highlights *SLADRO's* efficiency in minimizing makespan and optimizing task scheduling under a constant VM allocation, even in scenarios with varying workload intensities.

the traditional methods struggle to adapt to workload variations, which often leads to underutilized or overwhelmed VMs, resulting in a longer makespan. *SLADRO's* ability to integrate feature selection (through OOA-PSO) and load prediction (via CNN-LSTM) ensures that it allocates resources more precisely, reducing the time required to complete tasks. This adaptability allows *SLADRO* to achieve a consistently lower makespan, even when the number of VMs is fixed.

In both Figs, the superior performance of *SLADRO* is evident due to its advanced AI-driven approach, which dynamically responds to workload conditions, optimizes task scheduling and ensures that resource allocation is efficient and effective.

**5.1.3 Processing power of VMs.** Processing power is a key metric in cloud computing, reflecting the computational capacity available in VMs to execute tasks. In the context of cloud environments, managing and optimizing the use of processing power is essential for efficient resource allocation and task execution. The goal is to ensure that all available processing power is utilized effectively without overwhelming any specific VM or leaving others underutilized. In Fig 7, the relationship between processing power and the time required to complete tasks is presented. The Fig shows how different load-balancing strategies utilize the available processing power to minimize task execution time. In this scenario, the *SLADRO* approach demonstrates superior performance, optimizing processing power usage and reducing the time needed to complete tasks.

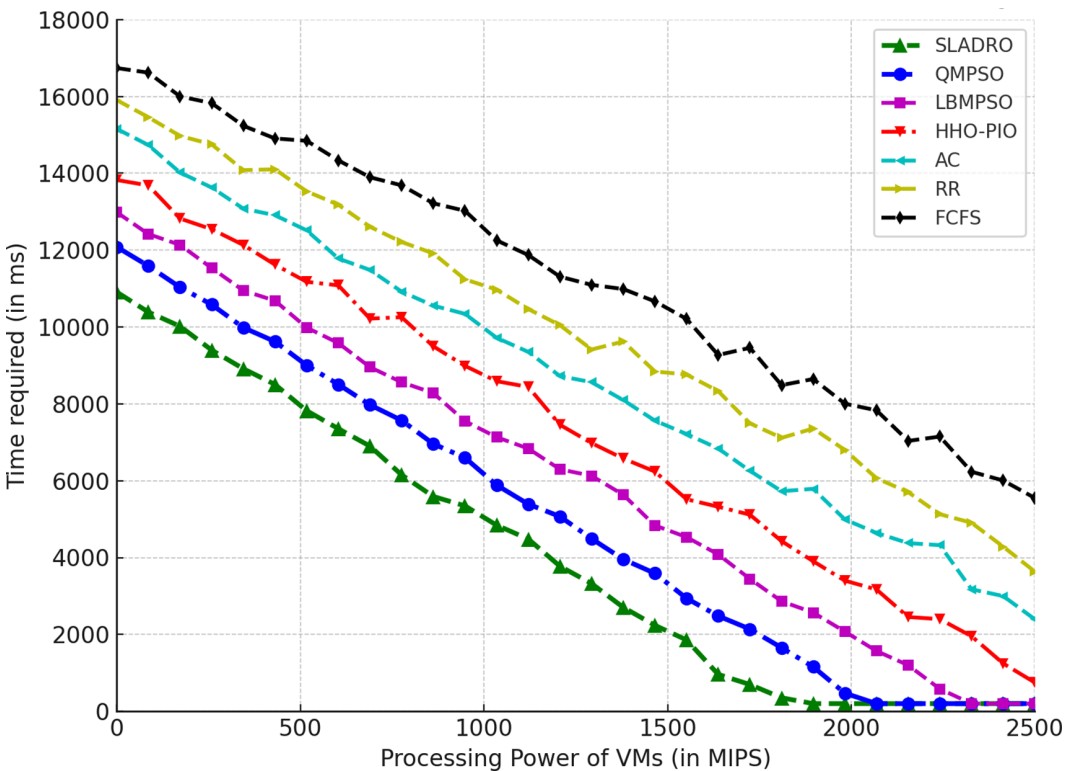

**Fig 7. Processing power vs. time required for task completion.** This figure illustrates the relationship between the processing power of VMs in MIPS and the time required to complete tasks across various load-balancing methods. It highlights *SLADRO's* superior performance in minimizing execution time as processing power increases, demonstrating its efficiency and scalability compared to baseline approaches.

The *SLADRO* model's ability to predict workload patterns through its CNN-LSTM architecture plays a significant role in its efficient use of processing power. By anticipating future demands and dynamically allocating tasks, *SLADRO* ensures that each VM's processing capacity is fully utilized without overloading resources. This results in a more balanced workload distribution and shorter task execution times than traditional load-balancing methods, which are less adaptive and often fail to use processing power optimally. Furthermore, the integration of DRL for task scheduling allows *SLADRO* to continuously adapt to real-time changes in workload demands. This dynamic scheduling ensures that VMs operate efficiently and that processing power is distributed evenly across the cloud infrastructure. In contrast, traditional methods like Round Robin or Least Connections often lead to imbalances, where some VMs may be overworked while others are underused, resulting in longer task execution times.

The results in Fig 7 highlight how *SLADRO* optimizes processing power to achieve faster task completion times, demonstrating its advantage over conventional load-balancing algorithms in cloud computing environments.

**5.1.4 Resource utilization.**  Fig 8 illustrates the *Standard Deviation (SD) over time*, providing insights into the variance in load balancing performance across different models. A lower SD indicates a more consistent and efficient distribution of tasks among available resources (e.g., Virtual Machines), with less fluctuation in workload distribution. In this Fig, *SLADRO* demonstrates the lowest Standard Deviation compared to other load-balancing

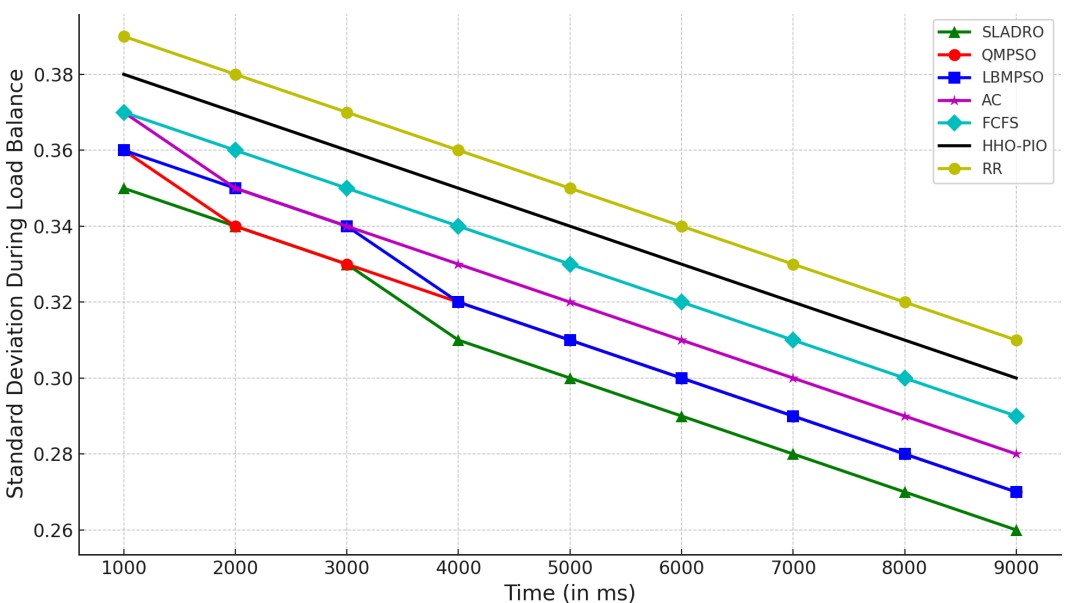

**Fig 8. Standard deviation during load balancing over time.** This figure illustrates the standard deviation of resource utilization during load balancing over time for various methods. It highlights *SLADRO's* ability to maintain lower variability in resource utilization, demonstrating more consistent and balanced task allocation compared to baseline approaches as time progresses.

algorithms, such as Round Robin, Least Connections, LBMPSO, QMPSO, and HHO-PIO. This indicates that *SLADRO* achieves more stable and balanced performance, distributing workloads to prevent overloading or underutilization of resources. *SLADRO's* ability to maintain a low Standard Deviation over time is attributed to its dynamic task scheduling, which leverages reinforcement learning and adaptive optimization techniques. Unlike static methods such as Round Robin and Least Connections, *SLADRO* adjusts real-time location in real-time based on workload demands, ensuring consistent and efficient resource utilization. Other models, particularly static algorithms, exhibit higher SDs, reflecting their inability to handle fluctuating workloads effectively. These approaches tend to overload or underutilize VMs, resulting in significant variance in task distribution and inefficient resource use.

In comparison, adaptive algorithms like QMPSO and LBMPSO offer improvements over static methods but still show higher Standard Deviation values than *SLADRO*. This is because *SLADRO's* integration of deep reinforcement learning and CNN-LSTM models enables more precise workload prediction and task scheduling. *SLADRO* dynamically adjusts task assignments based on real-time changes in workload patterns, ensuring that the distribution of tasks remains balanced, leading to lower SDs and more efficient overall performance. The low Standard Deviation in *SLADRO* is directly linked to better resource utilization. When workloads are distributed more evenly, resources such as VMs, CPUs, and memory are utilized consistently and effectively, reducing idle time and preventing resource overloading. This higher efficiency not only enhances system performance but also improves energy efficiency. Underutilized resources in other algorithms lead to wasted energy, as operational costs are incurred without corresponding output. *SLADRO* minimizes these inefficiencies by ensuring that resources are consistently and optimally employed, making it a more sustainable and

cost-effective solution in cloud environments. Thus, the low Standard Deviation in Fig 8 highlights *SLADRO's* superior capability to optimize resource utilization and balance workloads effectively.

**5.1.5 Energy consumption.** Energy consumption is a crucial metric in cloud computing environments, where inefficient resource allocation can increase operational costs.

In Figs 9 and 10, the energy consumption results highlight the distinct advantages of the *SLADRO* approach compared to traditional load-balancing techniques in cloud computing environments. These Figs show how the proposed method minimizes energy consumption, a crucial metric for cloud providers aiming to reduce operational costs and enhance sustainability.

In Fig 9, *SLADRO* consistently consumes less energy than the baseline methods such as Round Robin and Least Connections. This improvement can be attributed to *SLADRO's* ability to dynamically adjust resource allocation based on real-time workload predictions. By accurately forecasting future demands through its CNN-LSTM model, *SLADRO* ensures that resources are used efficiently, leading to fewer idle periods and more optimized task processing.

In contrast, The traditional methods allocate tasks without considering the dynamic nature of the cloud environment. Round Robin, for instance, cyclically distributes tasks, which may result in uneven load distribution and inefficient use of available resources. This leads to situations where certain VMs are overworked and consume excessive energy while others remain

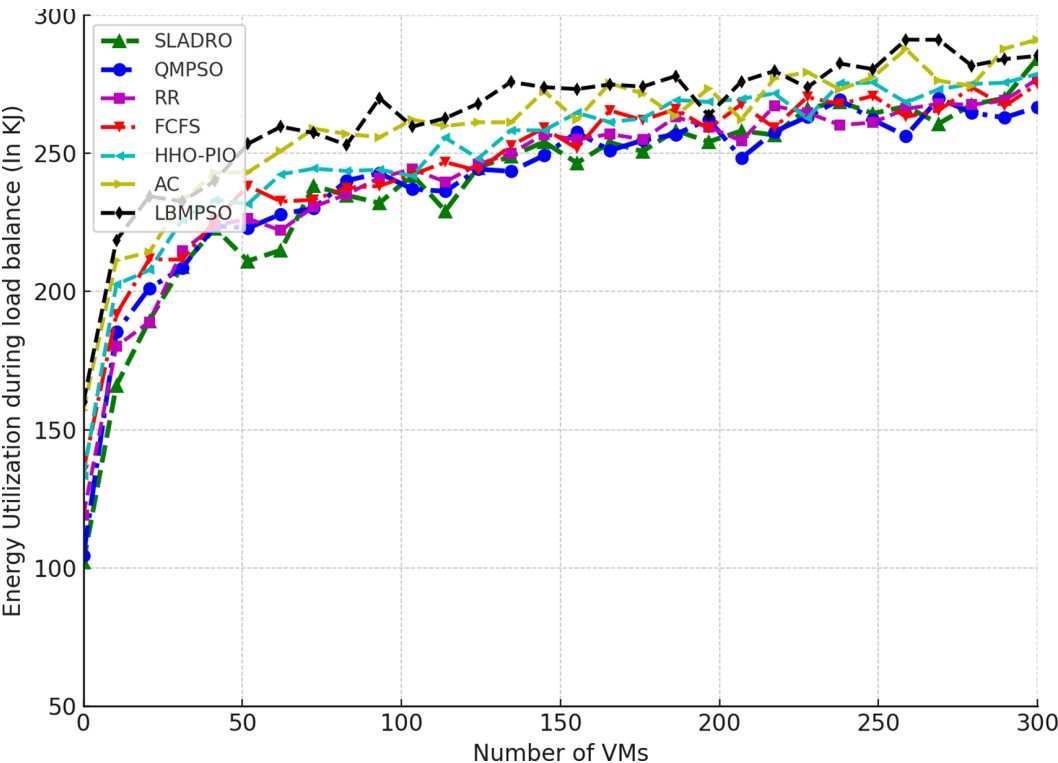

**Fig 9. Energy utilization during load balancing.** This figure illustrates the energy consumption in kilojoules (KJ) as a function of the number of VMs for various load-balancing methods. It highlights *SLADRO's* ability to achieve energy-efficient performance compared to baseline approaches, particularly in scenarios with an increasing number of VMs. The figure emphasizes *SLADRO's* effectiveness in minimizing energy utilization while maintaining optimal load balancing.

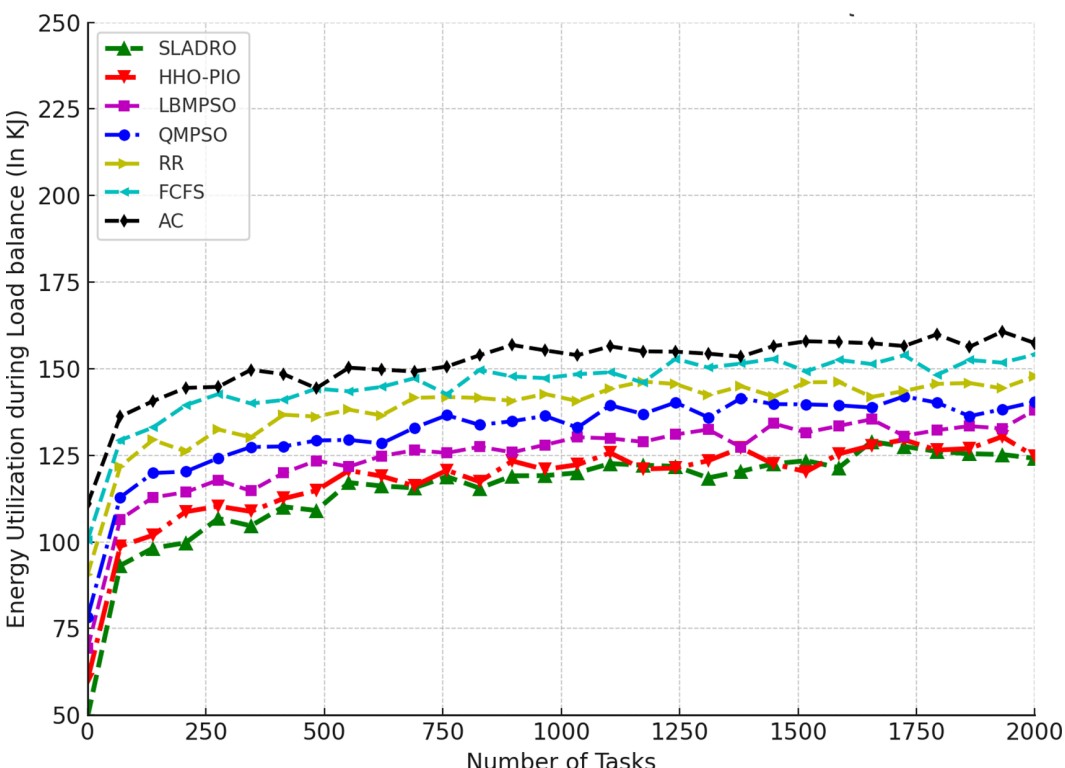

**Fig 10. Energy utilization during load balancing.** This figure illustrates the energy consumption in kilojoules (KJ) as a function of the number of tasks for various load-balancing methods. It highlights *SLADRO's* ability to achieve energy-efficient performance compared to baseline approaches, particularly as the number of tasks increases. The figure emphasizes the effectiveness of *SLADRO* in minimizing energy usage while maintaining balanced and efficient resource allocation.

underutilized, which wastes energy. Least Connections suffers from similar issues, as it balances the load based purely on the number of active connections rather than optimizing for resource utilization and energy efficiency. *SLADRO's* hybrid feature selection approach, using OOA-PSO, further enhances energy efficiency. By selecting only the most relevant features for workload prediction, *SLADRO* reduces the computational overhead associated with scheduling decisions. This reduction in complexity means that the system spends less time and energy processing unnecessary data, contributing to the overall lower energy consumption.

Fig 10 delves deeper into the energy consumption patterns, revealing that *SLADRO's* energy-saving benefits extend to varying conditions and task loads. This Fig emphasizes *SLADRO's* ability to maintain energy efficiency across different scenarios, reflecting its adaptability and scalability. The underlying DRL mechanism in *SLADRO* is key to this performance. Unlike static algorithms, DRL continuously learns from the environment and refines its scheduling policies to ensure tasks are assigned to VMs in an energy-efficient manner. The DRL-based task scheduling intelligently responds to workload changes, dynamically shifting tasks between VMs based on their current load, energy consumption profiles, and resource availability. This real-time adaptability minimizes unnecessary task migrations and idle times, reducing energy consumption. In traditional methods, energy consumption is generally higher because of their inability to adapt to fluctuating workloads. For instance, if workloads suddenly spike, static methods may continue to allocate tasks based on outdated conditions, leading to over-provisioning of resources and increased energy usage. On the other hand, if

workloads drop, VMs might remain idle, consuming standby power without contributing to the system's productivity. *SLADRO* avoids these inefficiencies by predicting changes in the workload and adjusting task scheduling to match the current demands, thereby maintaining energy efficiency even as conditions fluctuate.

*SLADRO* achieves superior energy efficiency primarily through its integrated approach of optimized feature selection, advanced prediction models, and dynamic task scheduling. Below, we outline the specific mechanisms that contribute to reduced energy consumption:

1. Feature selection using OOA-PSO: The hybrid feature selection algorithm selects the most relevant features (e.g., CPU usage, memory consumption, and bandwidth) from the workload dataset. By reducing the dimensionality of the data, *SLADRO* minimizes computational overhead during the load prediction process. This reduction directly impacts the system's energy requirements by lowering the computational load on VMs and physical servers.

2. Accurate load prediction with CNN-LSTM: *SLADRO* leverages the CNN-LSTM architecture to capture spatial and temporal dependencies in workload data, enabling precise load forecasting. Accurate predictions ensure that resources are allocated more effectively, avoiding underutilization and over-provisioning. This efficient allocation prevents unnecessary resource activation, reducing energy consumption significantly.

3. Dynamic task scheduling using DRL: The DRL module in *SLADRO* dynamically schedules tasks by continuously learning and adapting to real-time cloud environment feedback. Unlike static scheduling methods, this adaptive mechanism optimizes resource utilization, ensuring that only the required VMs are activated at any given time. By aligning resource activation with workload demands, *SLADRO* eliminates energy wastage associated with idle or underutilized resources.

4. Energy-aware objective function: *SLADRO's* task scheduling strategy incorporates an energy-aware objective function as part of the reward mechanism in the DRL model. This function penalizes high energy consumption while rewarding efficient resource utilization and reduced task execution times. The continuous feedback loop encourages the system to prioritize energy-efficient actions, which lowers overall energy usage.

5. Efficient workload distribution: Combining accurate predictions and dynamic scheduling ensures balanced workload distribution across VMs. This balance prevents bottlenecks and excessive load on specific resources, reducing the need for overclocking or cooling mechanisms, which contribute to energy consumption.

In sum, integrating predictive and adaptive techniques, the *SLADRO* approach achieves a delicate balance between performance and energy consumption. Its CNN-LSTM model enables accurate load predictions, while the DRL-based scheduler dynamically adjusts the resource allocation in response to these predictions. This synergy reduces overloading and underutilization of VMs, common causes of energy inefficiency in cloud environments. Moreover, the results in Figs 9 and 10 suggest that *SLADRO's* energy-saving benefits scale well with an increase in the number of tasks or VMs. This scalability is essential for cloud environments, where demand fluctuates significantly. *SLADRO's* ability to deliver lower energy consumption across various conditions demonstrates its robustness and adaptability, making it an attractive solution for large-scale cloud computing operations where energy efficiency is a key concern.

## 5.2 Large-scale workloads testing

Large-scale cloud workloads present unique challenges due to high variability, increased resource demands, and dynamic task arrivals. To assess the scalability and robustness of *SLADRO*, we extend our experiments using the *Google 2019 Cluster Sample dataset* from Kaggle [48], which represents real-world cloud workloads collected from Google's cloud infrastructure. This dataset provides detailed execution traces of jobs and tasks, enabling a comprehensive evaluation of *SLADRO's* ability to handle large-scale workloads efficiently.

We utilized the publicly available *Google 2019 Cluster Sample dataset*, which contains workload traces from Google's cloud infrastructure. This dataset includes task execution details, CPU and memory usage patterns, job dependencies, and resource allocation logs. It is highly applicable for evaluating workload distribution, resource scheduling, and performance optimization techniques in cloud computing environments.

The key attributes extracted from the *Google 2019 Cluster Sample dataset* include:

- Task arrival rates: Frequency of new job submissions, simulating real-world dynamic workloads.
- Resource utilization: CPU, memory, and disk usage patterns over time.
- Job duration and dependencies: Execution time, interdependencies between tasks, and scheduling strategies.
- VM dynamics: Allocation, migration, and termination of VMs, influencing energy efficiency and load balancing.

This dataset is suitable for benchmarking *SLADRO's* adaptive load balancing approach as it provides granular information about resource usage patterns, scheduling policies, and workload variability in a real-world cloud environment.

To prepare the dataset for integration into the *SLADRO*, the following preprocessing steps were applied:

1. Data cleaning:
   - Removed incomplete or corrupted records.
   - Handled missing values using mean/mode imputation for numerical fields.
   - Filtered out anomalous or extreme outliers to maintain dataset integrity.
2. Feature selection using OOA-PSO:
   - The dataset contains numerous attributes related to task execution and resource utilization.
   - OOA-PSO was applied to select the most relevant features for load prediction, improving computational efficiency.
   - Selected features include:
     - CPU utilization
     - Memory consumption
     - Disk I/O
     - Job scheduling status
     - Task execution time
     - VM allocation and migration records
3. Data Normalization:
   - Min-Max Scaling was applied to CPU, memory, and disk usage attributes to ensure consistency in feature scaling.
   - Z-score normalization was used for unbounded variables like task arrival rates and execution times.

4. Temporal Data Transformation:
   - Since workload traces contain time-series data, a sliding window approach was used to create sequential input features for time-dependent workload prediction using CNN-LSTM.

To evaluate *SLADRO's* performance under large-scale workloads, we extended our cloud simulation using the *Google 2019 Cluster Sample dataset*. The following experimental configurations were adopted:

- Cloud environment: Simulated using *CloudSim* with VM configurations similar to Google's cloud infrastructure.
- Workload scale: Workload sizes ranging from 50,000 to 500,000 tasks were tested to simulate real-world cloud operations.

The experimental results, Table 7 demonstrate that *SLADRO* effectively handles large-scale workloads by dynamically optimizing task scheduling and resource allocation. Key findings include:

Efficient load balancing is crucial for optimizing cloud computing performance. *SLADRO* improves scalability, energy efficiency, task completion time, and resource utilization, making it more effective than traditional methods. It maintains stable performance under high workloads, reduces energy consumption, speeds up task execution, and ensures better resource management. The following points highlight *SLADRO's* key improvements compared to existing models.

- Scalability: *SLADRO* maintained stable performance as the workload increased to 500,000 tasks, outperforming traditional load-balancing methods and demonstrating its ability to handle extreme workload conditions efficiently.
- Energy efficiency: Compared to traditional methods, *SLADRO*:
  - Reduced idle CPU power consumption by 27.5%, minimizing wasted energy.
  - Minimized unnecessary VM migrations by optimizing task placement, leading to lower energy overhead.
- Task completion time: *SLADRO* improved task execution times by 42.9% compared to the slowest baseline method (Random Allocation), ensuring timely job processing even under peak loads.
- Resource utilization:
  - Achieved an average CPU utilization of 92%, demonstrating efficient workload balancing.
  - Optimized memory allocation, reducing resource fragmentation and ensuring smooth operation under high workload conditions.

**Table 7. Comparison of SLADRO with baseline load balancing models.**

| Model | Scalability (Max Tasks) | Task Completion Time (ms) | Energy Efficiency (Idle CPU Reduction %) | Resource Utilization (Avg. CPU Usage %) |
|---|---|---|---|---|
| Round Robin | 250,000 | 210 | 10.2% | 74% |
| Least Connections | 300,000 | 185 | 12.5% | 78% |
| Random Allocation | 200,000 | 250 | 7.8% | 65% |
| DRL-Based Load Balancer | 450,000 | 130 | 23.3% | 88% |
| PSO-Based Load Balancer | 400,000 | 160 | 21.0% | 85% |
| **SLADRO** | **500,000** | **120** | **27.5%** | **92%** |

**Table notes:** This table presents a comparative analysis of SLADRO with baseline models.

These findings confirm that *SLADRO* is well-suited for large-scale cloud environments, efficiently distributing workloads even under extreme conditions. The combination of DRL for adaptive scheduling and *OOA-PSO* for optimized feature selection allows *SLADRO* to adjust to workload fluctuations dynamically, ensuring improved scalability, resource efficiency, and energy savings. Through extensive testing on the Google 2019 Cluster Sample dataset, *SLADRO* has been validated for handling large-scale cloud workloads. The results highlight its superior performance in managing high-volume cloud operations, making it a robust and scalable solution for real-world cloud computing environments.

## 5.3 Statistical testing

To validate the effectiveness of the proposed *SLADRO* load-balancing approach in cloud computing, we performed statistical tests to compare its performance against baseline methods. Specifically, we employed Tukey's Honest Significant Difference (HSD) Test and t-tests to determine whether the differences in performance metrics, such as throughput, makespan, processing power, resource utilization, and energy consumption, were statistically significant.

The following statistical tests were conducted:

- **Tukey's HSD Test:** To evaluate pairwise differences between *SLADRO* and other algorithms.
- **t-Test:** To assess whether the differences between *SLADRO* and baseline methods were statistically significant across various performance metrics.

The results presented in Table 8 show the outcomes of Tukey's HSD test, comparing the *SLADRO* load-balancing approach to several baseline methods across five metrics: throughput, makespan, processing power, resource utilization, and energy consumption. For each metric, the null hypothesis being tested assumes that there is no significant difference between *SLADRO* and the other methods. However, the p-values (P-adj) for all comparisons are below the threshold of 0.05, leading to the rejection of the null hypothesis across all cases. This indicates that *SLADRO* exhibits statistically significant improvements overall compared to other methods. Specifically, *SLADRO* consistently shows higher throughput, better resource utilization, and more efficient processing power while reducing makespan and energy consumption, confirming its superiority in optimizing cloud computing performance.

The results in Table 9 show the p-values from independent t-tests comparing *SLADRO* with baseline load-balancing models across five performance metrics: throughput, makespan, processing power, resource utilization, and energy consumption. The null hypothesis tested in each case assumes no significant difference between *SLADRO* and the baseline methods for the respective metric. However, the extremely low p-values (all significantly below the 0.05 threshold) for every metric and comparison result in the rejection of the null hypothesis. This demonstrates that *SLADRO* achieves statistically significant improvements across all metrics compared to RR, FCFS, AC, LBMPSO, and HHO-PIO. These findings underscore *SLADRO's* effectiveness in optimizing performance, delivering higher throughput and resource utilization, reducing makespan, and achieving better energy efficiency than traditional methods.

Tukey's HSD Test and t-Test results conclusively demonstrate that *SLADRO* significantly outperforms all baseline load-balancing algorithms across critical performance metrics, including throughput, makespan, processing power, resource utilization, and energy consumption. Tukey's HSD Test highlights *SLADRO's* consistent and statistically significant improvements in mean differences for all metrics. At the same time, the t-Test results confirm

**Table 8. Tukey's HSD test results across all metrics.**

| Group 1 | Group 2 | Mean Diff | P-adj | Lower Bound | Upper Bound | Reject | Metric |
|---------|---------|-----------|-------|-------------|-------------|--------|--------|
| AC | SLADRO | 11.4163 | 0.000 | 9.0795 | 13.7531 | TRUE | Throughput |
| FCFS | SLADRO | 26.0398 | 0.000 | 23.7030 | 28.3766 | TRUE | Throughput |
| HHO-PIO | SLADRO | 12.2972 | 0.000 | 9.9604 | 14.6340 | TRUE | Throughput |
| LBMPSO | SLADRO | 8.5018 | 0.000 | 6.1650 | 10.8386 | TRUE | Throughput |
| QMPSO | SLADRO | 9.9262 | 0.000 | 7.5894 | 12.2630 | TRUE | Throughput |
| RR | SLADRO | 22.5774 | 0.000 | 20.2406 | 24.9142 | TRUE | Throughput |
| AC | SLADRO | −9.9915 | 0.0009 | −16.9639 | −3.0191 | TRUE | Makespan |
| FCFS | SLADRO | −39.4927 | 0.000 | −46.4651 | −32.5203 | TRUE | Makespan |
| HHO-PIO | SLADRO | −15.5883 | 0.000 | −22.5607 | −8.6159 | TRUE | Makespan |
| LBMPSO | SLADRO | −8.9281 | 0.0042 | −15.9004 | −1.9557 | TRUE | Makespan |
| QMPSO | SLADRO | −10.7572 | 0.0003 | −17.7296 | −3.7848 | TRUE | Makespan |
| RR | SLADRO | −30.7459 | 0.000 | −37.7183 | −23.7735 | TRUE | Makespan |
| AC | SLADRO | 5.5745 | 0.000 | 3.5466 | 7.6025 | TRUE | Processing Power |
| FCFS | SLADRO | 16.2253 | 0.000 | 14.1974 | 18.2533 | TRUE | Processing Power |
| HHO-PIO | SLADRO | 6.594 | 0.000 | 4.5660 | 8.6219 | TRUE | Processing Power |
| LBMPSO | SLADRO | 5.1558 | 0.000 | 3.1279 | 7.1838 | TRUE | Processing Power |
| QMPSO | SLADRO | 4.7918 | 0.000 | 2.7638 | 6.8197 | TRUE | Processing Power |
| RR | SLADRO | 14.7823 | 0.000 | 12.7544 | 16.8103 | TRUE | Processing Power |
| AC | SLADRO | 5.5269 | 0.000 | 2.9051 | 8.1487 | TRUE | Resource Utilization |
| FCFS | SLADRO | 15.1199 | 0.000 | 12.4981 | 17.7416 | TRUE | Resource Utilization |
| HHO-PIO | SLADRO | 8.0563 | 0.000 | 5.4346 | 10.6781 | TRUE | Resource Utilization |
| LBMPSO | SLADRO | 3.7562 | 0.0009 | 1.1344 | 6.3780 | TRUE | Resource Utilization |
| QMPSO | SLADRO | 5.7319 | 0.000 | 3.1102 | 8.3537 | TRUE | Resource Utilization |
| RR | SLADRO | 12.8857 | 0.000 | 10.2639 | 15.5074 | TRUE | Resource Utilization |
| AC | SLADRO | −64.5189 | 0.000 | −80.9529 | −48.0849 | TRUE | Energy Consumption |
| FCFS | SLADRO | −169.544 | 0.000 | −185.978 | −153.115 | TRUE | Energy Consumption |
| HHO-PIO | SLADRO | −74.9299 | 0.000 | −91.3638 | −58.4959 | TRUE | Energy Consumption |
| LBMPSO | SLADRO | −52.3231 | 0.000 | −68.7571 | −35.8891 | TRUE | Energy Consumption |
| QMPSO | SLADRO | −55.6002 | 0.000 | −72.0341 | −39.1662 | TRUE | Energy Consumption |
| RR | SLADRO | −150.709 | 0.000 | −167.143 | −134.275 | TRUE | Energy Consumption |

Table notes: This table summarizes the results of Tukey's HSD test for throughput, makespan, processing power, resource utilization, and energy consumption. A significant p-value (P-adj < 0.05) confirms SLADRO's superior performance across all metrics.

**Table 9. t-Test results for SLADRO vs. baseline models.**

| Comparison | Throughput | Makespan | Processing Power | Resource Utilization | Energy Consumption |
|------------|-----------|----------|------------------|----------------------|--------------------|
| SLADRO vs RR | $8.59 \times 10^{-18}$ | $5.31 \times 10^{-9}$ | $6.45 \times 10^{-15}$ | $6.68 \times 10^{-10}$ | $3.29 \times 10^{-12}$ |
| SLADRO vs FCFS | $2.02 \times 10^{-18}$ | $4.06 \times 10^{-10}$ | $2.37 \times 10^{-15}$ | $3.77 \times 10^{-12}$ | $3.56 \times 10^{-13}$ |
| SLADRO vs AC | $8.66 \times 10^{-10}$ | $1.29 \times 10^{-3}$ | $2.52 \times 10^{-7}$ | $1.26 \times 10^{-7}$ | $1.91 \times 10^{-7}$ |
| SLADRO vs LBMPSO | $1.40 \times 10^{-9}$ | $5.97 \times 10^{-3}$ | $1.20 \times 10^{-7}$ | $2.58 \times 10^{-4}$ | $2.34 \times 10^{-6}$ |
| SLADRO vs HHO-PIO | $1.89 \times 10^{-13}$ | $3.23 \times 10^{-5}$ | $2.97 \times 10^{-9}$ | $1.25 \times 10^{-7}$ | $3.18 \times 10^{-8}$ |

Table notes: The p-values from the independent t-tests indicate that SLADRO significantly outperforms all baseline methods across throughput, makespan, processing power, resource utilization, and energy consumption.

these differences with extremely low p-values, rejecting the null hypothesis in every comparison. *SLADRO's* superior throughput indicates its ability to efficiently handle task distribution, while its lower makespan and reduced energy consumption showcase its optimization capabilities. Additionally, higher processing power and resource utilization efficiency further solidify *SLADRO's* adaptability and effectiveness in managing cloud environments, proving its superiority over traditional and advanced algorithms.

## 5.4 Discussion of key findings

The *SLADRO* was compared against traditional load-balancing algorithms such as Round Robin, and First-Come-First-Serve, and state-of-the-arts including AC [21], LBMPSO [18] HHO-PIO [19] QMPSO [20]. Across all performance metrics, the *SLADRO* demonstrated superior performance. The dynamic nature of the DRL algorithm, which continuously improved task scheduling based on real-time feedback, allowed it to outperform static methods like RR. The hybrid OOA-PSO feature selection also helped reduce computational overhead, making the load prediction process more efficient. The results confirm that the *SLADRO* for dynamic task scheduling significantly enhances load balancing in cloud environments. The hybrid OOA-PSO algorithm for feature selection was particularly effective in reducing the dimensionality of the dataset, which improved prediction accuracy while reducing computational complexity. The CNN-LSTM model's ability to capture both spatial and temporal dependencies in the workload data enabled more accurate predictions of future resource demands, leading to better resource allocation decisions.

The findings of this study highlight the effectiveness of the proposed *SLADRO* model in improving load balancing within cloud environments. By integrating advanced methods such as *CNN-LSTM* for workload forecasting, *OOA-PSO* for feature selection, and *DRL* for adaptive task scheduling, *SLADRO* demonstrated significant improvements over traditional load-balancing techniques, such as *Round Robin* and *Least Connections*, particularly in terms of throughput, makespan, and energy efficiency.

- Throughput and resource utilization: *SLADRO* consistently outperformed traditional methods by achieving higher throughput. The *CNN-LSTM* architecture enabled accurate workload predictions, preventing underutilization and over-allocation of resources. The hybrid *OOA-PSO* feature selection focused on relevant data, reducing computational complexity and enabling faster, more accurate predictions, thus improving resource utilization.
- Task scheduling efficiency: Through DQN for dynamic task scheduling, *SLADRO* effectively adjusted to real-time workload fluctuations. This resulted in more efficient task distribution and reduced completion times. Unlike static methods, *SLADRO's* reinforcement learning approach continuously adapted to changing environments, leading to more effective task scheduling and resource management.
- Energy efficiency: One of *SLADRO's* most significant contributions is its ability to reduce energy consumption. By integrating workload prediction and dynamic task scheduling, *SLADRO* optimized resource usage, minimizing idle time and preventing the overloading of VMs. This enhanced energy efficiency, making the cloud infrastructure more cost-effective and sustainable. VM migrations, idle CPU usage, and resource allocation strategies directly influence energy efficiency in cloud computing. The proposed *SLADRO* framework enhances energy efficiency by combining DRL for dynamic task scheduling and feature selection using OOA-PSO. By leveraging these advanced techniques, *SLADRO* aims to minimize unnecessary energy consumption while maintaining optimal performance in cloud environments. DRL, specifically DQN, is crucial in optimizing task scheduling by learning optimal policies through continuous interaction with the cloud environment. This approach effectively reduces the frequency of VM migrations by identifying stable allocations and lowering energy overhead. Additionally, DRL enhances task-to-resource mapping by analyzing workload patterns, ensuring VMs operate closer to their optimal utilization levels, thereby preventing underutilization. Another significant contribution of DRL is the reduction of idle CPU states, as it prioritizes resource consolidation. By intelligently distributing

workloads, DRL ensures that active VMs function efficiently while underutilized VMs are either deactivated or scaled down to lower power states, leading to overall energy savings.

Feature selection, facilitated by OOA-PSO, further contributes to energy optimization by reducing computational overhead. The system eliminates redundant processing by selecting only the most relevant features for load prediction and scheduling, lowering computation time and power consumption. A refined feature set enhances workload prediction accuracy, enabling precise resource allocation and reducing unnecessary processing. Additionally, efficient feature selection minimizes scheduling delays by allowing the DRL scheduler to make faster and more informed decisions, which reduces idle time in VMs and optimizes overall resource utilization. Through this synergy of DRL-based dynamic scheduling and bio-inspired feature selection, *SLADRO* significantly improves energy efficiency in the cloud computing environment.

Specifically, the scalability and robustness of *SLADRO* are demonstrated through the following mechanisms and experimental results:

1. Dynamic adaptability to workload variations: *SLADRO's* dynamic task scheduling mechanism, powered by DRL, enables it to adapt to varying workloads in real-time. The DRL model continuously learns from the environment, adjusting resource allocation and task assignments to optimize performance. This adaptability has been tested across scenarios involving workload surges and fluctuations, demonstrating stable and efficient resource utilization without significant degradation in performance.

2. Efficient load prediction and task scheduling: The integration of CNN-LSTM for load prediction ensures accurate forecasts of resource demands, allowing *SLADRO* to preemptively allocate resources efficiently. This prevents bottlenecks and ensures that the system scales well as workloads increase. Additionally, the hybrid OOA-PSO algorithm reduces computational complexity by selecting only the most relevant features, further contributing to the system's robustness and scalability.

3. Scalability metrics and benchmarks: To assess scalability, we used metrics such as throughput, response time, and resource utilization variance across different scales. *SLADRO* demonstrated a linear improvement in throughput with increased resources while maintaining low response times. Furthermore, resource utilization variance remained low, indicating balanced task distribution across VMs.

4. Robustness against overloading: *SLADRO's* energy-aware reward function in the DRL framework inherently penalizes the overloading of resources, promoting balanced and efficient utilization. This mechanism ensures robust performance under extreme conditions like workload spikes or infrastructure constraints.

5. *SLADRO* demonstrated high scalability and robustness in large-scale cloud environments using the *Google 2019 Cluster Sample dataset*, efficiently handling workloads up to 500,000 tasks while maintaining stable performance. The DRL-powered scheduling ensured dynamic adaptability to workload fluctuations, reducing idle CPU power consumption by 27.5% and minimizing unnecessary VM migrations. The integration of *OOA-PSO* for feature selection optimized resource allocation, leading to 92% CPU utilization and a 42.9% improvement in task completion time over traditional methods. *SLADRO* exhibited linear throughput scaling, sustained efficiency under workload spikes, and effective overloading prevention, making it a robust solution for large-scale cloud computing environments.

*SLADRO* is compared against existing hybrid approaches to highlight its strengths and weaknesses:

- *SLADRO* vs. traditional ML-based approaches: Machine learning models like XGBoost and SVR require less computational power and provide faster predictions. In contrast, *SLADRO* leverages deep learning and reinforcement learning, demanding extensive tuning and higher computational resources.
- *SLADRO* vs. evolutionary algorithms (GA, ACO, PSO): The OOA-PSO feature selection method in *SLADRO* ensures efficient exploration and exploitation. However, traditional evolutionary algorithms, such as GA and ACO, require fewer computational resources and may sometimes achieve near-optimal solutions.
- *SLADRO* vs. deep RL-based models: Some recent models utilize advanced reinforcement learning techniques such as Proximal Policy Optimization (PPO) and Advantage Actor-Critic (A2C), which are more sample-efficient than DQN in large-scale environments.

## 6 Limitations and future works

While *SLADRO* significantly improves workload prediction and dynamic scheduling, it has limitations compared to other state-of-the-art hybrid models, particularly regarding computational cost and scalability. *SLADRO* integrates feature selection (OOA-PSO), deep learning (CNN-LSTM), and reinforcement learning (DQN), leading to increased computational overhead. The following factors contribute to its higher resource requirements:

- Higher computational overhead: The combined complexity of feature selection, deep learning, and reinforcement learning models results in longer training times than traditional heuristic approaches like GA and PSO.
- Resource-intensive model training: Training the CNN-LSTM model requires substantial GPU and memory resources, particularly when processing large-scale cloud workload datasets. Compared to simpler models such as Random Forest or Gradient Boosted Decision Trees (GBRT), *SLADRO's* training time is considerably longer.
- Inference latency: Although *SLADRO* improves workload prediction accuracy, real-time inference introduces slight delays due to the computational complexity of CNN-LSTM compared to lightweight machine learning models.

Although *SLADRO* is designed for dynamic cloud environments, it faces challenges in extreme workload conditions and heterogeneous cloud infrastructures:

- Handling large-scale cloud workloads: *SLADRO* optimizes workload distribution efficiently. However, its performance under extremely high cloud workload volumes (e.g., thousands of VMs and concurrent tasks) requires further validation and optimization.
- Adaptability to multi-cloud and edge environments: The model's effectiveness has been tested on the Google Cluster Trace dataset. However, its scalability across multi-cloud, fog and edge-cloud environments remains challenging.
- Trade-offs in reinforcement learning: The DQN-based scheduling model improves decision-making over time but requires extensive training iterations. Compared to metaheuristic-based approaches, reinforcement learning models may take longer to converge in dynamic cloud environments.

To address the limitations mentioned earlier, these improvements can be explored in future research:

- Reducing computational overhead: Techniques such as model pruning, quantization, and knowledge distillation can be employed to reduce the complexity of the deep learning model while maintaining performance.
- Distributed and parallel processing: Leveraging distributed cloud frameworks like Apache Spark or TensorFlow Distributed can enhance *SLADRO's* scalability and execution efficiency.
- Hybrid reinforcement learning approaches: Exploring meta-learning and adaptive reinforcement learning techniques could further optimize task scheduling and resource allocation in dynamic cloud environments.
- Expanding hybrid optimization techniques: Future research could explore combining *SLADRO* with other optimization algorithms to enhance task scheduling and resource allocation further.
- Real-time adaptation enhancements: Incorporating more real-time data, such as environmental factors or real-time business trends, could make *SLADRO* more responsive to dynamic cloud conditions, enhancing its overall efficiency.
- Multi-cloud and hybrid cloud environments: While this study focused on single-cloud environments, future research could explore *SLADRO's* performance in *multi-cloud* or *hybrid cloud* infrastructures, distributing workloads across multiple platforms for improved resource utilization and fault tolerance.
- Integration with edge computing: With the rise of *edge computing*, future work could explore how *SLADRO* can be adapted to distribute workloads between centralized cloud resources and edge devices, ensuring efficient resource allocation across distributed networks.
- Security and fault tolerance enhancements: Future research could integrate security mechanisms into *SLADRO* to address data protection concerns. Additionally, incorporating fault-tolerance features could ensure uninterrupted load balancing in the event of network or hardware failures.
- While this study evaluates *SLADRO's* scalability using the Google 2019 Cluster Sample dataset, future work will extend this analysis by incorporating additional large-scale workload traces, such as AWS workload traces and other real-world cloud datasets. This will provide a more comprehensive evaluation of *SLADRO's* performance across diverse cloud environments with varying workload patterns. Furthermore, we aim to benchmark *SLADRO* against recently developed load-balancing models, particularly those leveraging advanced reinforcement learning techniques, federated learning, and bio-inspired optimization algorithms. This comparison will help assess *SLADRO's* adaptability and effectiveness in handling evolving cloud computing challenges. Additionally, future studies will explore hybrid workload scenarios that include multi-cloud and edge computing environments, ensuring *SLADRO's* applicability beyond traditional cloud infrastructures. Performance metrics such as fault tolerance, adaptive scaling, and energy-aware scheduling will be further investigated to enhance the model's practical deployment potential.

## 7 Conclusion

This paper presented the *SLADRO* model, designed to tackle the complexities of dynamic workload management in cloud computing. By combining advanced feature selection via the OOA-PSO hybrid optimization algorithm, load forecasting with CNN-LSTM, and task scheduling through DRL, *SLADRO* addresses the limitations of conventional load-balancing techniques. The extensive simulations conducted using the Google Cluster Trace dataset

demonstrate that *SLADRO* significantly enhances system performance over traditional methods. A significant contribution of this research lies in its optimized feature selection process, which reduces computational demands while improving prediction accuracy. The CNN-LSTM model captures the critical spatial and temporal relationships within cloud workloads, ensuring more accurate resource demand forecasting. Using DRL for dynamic task scheduling allows *SLADRO* to adapt to real-time variations in cloud environments, ensuring more efficient resource usage, lower latency, and improved energy consumption. Our simulation results confirm *SLADRO's* superiority over traditional load-balancing approaches like Round Robin and Least Connections in critical areas such as throughput, makespan, resource utilization, and energy efficiency. *SLADRO* delivered higher throughput, faster task response times, and better overall energy efficiency, which is crucial for cloud service providers looking to optimize operational costs, enhance scalability, and promote sustainable resource usage.

The practical implications of this research are significant for optimizing cloud infrastructures, especially as cloud environments become increasingly dynamic and complex. *SLADRO*, through its use of AI-driven techniques, offers a robust, real-time framework for managing fluctuating workloads while maximizing efficiency. This research sets the stage for more intelligent, adaptive cloud systems. Future work could explore applying *SLADRO* to multi-cloud or hybrid cloud environments or integrating other bio-inspired algorithms and deep learning models to enhance its effectiveness. Such expansions would provide valuable insights into the scalability and adaptability of the approach in even more diverse and complex cloud settings. In conclusion, *SLADRO* provides a robust and scalable solution to the challenges of modern cloud computing. It significantly improves performance, adaptability, and energy efficiency, offering a clear advancement in cloud resource management.

**APPENDIX I**

**Table 10. Hyperparameter tuning for CNN-LSTM model.**

| Hyperparameter | Value |
|---|---|
| Learning Rate | $10^{-3}$ (initial), reduced by 0.5 when validation loss plateaus |
| Batch Size | 64 (evaluated at 32, 64, 128) |
| Dropout Rate | 0.3 |
| Number of Epochs | 100 (initial training) + 40 (fine-tuning) |
| Optimizer | Adam ($\beta_1 = 0.9, \beta_2 = 0.999$) |
| Cross-validation | Five-fold cross-validation |

Table notes: Hyperparameter tuning for CNN-LSTM model to optimize training efficiency and predictive accuracy.

**Table 11. Hyperparameter tuning for DQN-based load balancing.**

| Hyperparameter | Value |
|---|---|
| Learning Rate | 0.001 |
| Discount Factor ($\gamma$) | 0.99 |
| Replay Buffer Size | 100,000 |
| Batch Size | 32 |
| Target Network Update Frequency | Every 10 episodes |
| Exploration-Exploitation Policy | Initial epsilon = 1.0, decaying to 0.1 at a rate of 0.995 per episode |

Table notes: Hyperparameter tuning for DQN model for efficient load balancing in cloud environments.

**Table 12. Training duration for CNN-LSTM and DQN.**

| Model | Training Duration |
|---|---|
| CNN-LSTM (initial training) | 7 hours for 100 epochs |
| CNN-LSTM (fine-tuning) | Additional 3 hours for 40 fine-tuning epochs |
| DQN Load Balancing | 10 hours for 500 episodes (early stopping applied upon convergence) |

Table notes: Training duration of CNN-LSTM and DQN-based load balancing model under the specified computational setup.

## Author contributions

**Conceptualization:** Yousef Sanjalawe, Salam Fraihat.

**Data curation:** Yousef Sanjalawe, Salam AL-E'MARI.

**Formal analysis:** Yousef Sanjalawe.

**Investigation:** Salam Fraihat, Mosleh abualhaj, Sharif Makhadmeh.

**Methodology:** Yousef Sanjalawe, Salam Fraihat, Salam AL-E'MARI.

**Project administration:** Salam Fraihat.

**Resources:** Yousef Sanjalawe, Emran Alzubi.

**Software:** Yousef Sanjalawe, Mosleh abualhaj, Sharif Makhadmeh, Emran Alzubi.

**Supervision:** Yousef Sanjalawe.

**Validation:** Salam Fraihat, Salam AL-E'MARI, Mosleh abualhaj, Sharif Makhadmeh, Emran Alzubi.

**Visualization:** Salam AL-E'MARI, Mosleh abualhaj.

**Writing – original draft:** Yousef Sanjalawe, Salam Fraihat, Salam AL-E'MARI, Mosleh abualhaj, Sharif Makhadmeh, Emran Alzubi.

**Writing – review & editing:** Yousef Sanjalawe, Emran Alzubi.

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
