## [Decision Letter · Decision Letter 0]

27 Dec 2024

PONE-D-24-52330Smart Load Balancing in Cloud Computing: Integrating Feature Selection with Advanced Deep Learning ModelsPLOS ONE

Dear Dr. Fraihat,

Thank you for submitting your manuscript to PLOS ONE. After careful consideration, we feel that it has merit but does not fully meet PLOS ONE’s publication criteria as it currently stands. Therefore, we invite you to submit a revised version of the manuscript that addresses the points raised during the review process.

When preparing a revised version of your manuscript, please address all comments raised by the referees.

We look forward to receiving your revised manuscript.

Kind regards,

Davide La Torre

Academic Editor

PLOS ONE

Journal Requirements:

3. Please ensure that you refer to Figure 3 in your text as, if accepted, production will need this reference to link the reader to the figure.

4. We are unable to open your Supporting Information file plos2015.bst and plos_latex_template.tex. Please kindly revise as necessary and re-upload.

Additional Editor Comments :

The paper is promising with interesting findings. However, revisions are needed to address the referees' comments, which will enhance clarity and strengthen the analysis. I encourage the authors to consider these suggestions when revising their manuscript.

Reviewers' comments:

Reviewer's Responses to Questions

**Comments to the Author**

1. Is the manuscript technically sound, and do the data support the conclusions?

Reviewer #1: Yes

Reviewer #2: Yes

2. Has the statistical analysis been performed appropriately and rigorously? 

Reviewer #1: Yes

Reviewer #2: Yes

3. Have the authors made all data underlying the findings in their manuscript fully available?

Reviewer #1: Yes

Reviewer #2: Yes

4. Is the manuscript presented in an intelligible fashion and written in standard English?

Reviewer #1: Yes

Reviewer #2: No

5. Review Comments to the Author

Reviewer #1: The paper addresses a significant issue in cloud computing resource management by proposing a smart load-balancing strategy that integrates advanced techniques to overcome the limitations of traditional methods. The proposed SLADRO approach incorporates Convolutional Neural Networks (CNN) and Long Short-Term Memory (LSTM) networks for load prediction, Orthogonal Arrays and Particle Swarm Optimization (OOA-PSO) for feature selection, and Deep Reinforcement Learning (DRL) for dynamic task scheduling, aiming to improve resource utilization and energy efficiency.

While the proposed methodology is promising, I have several concerns and suggestions that I believe can enhance the clarity, applicability, and overall quality of the paper:

1- In Section 3.1.2, the authors use both Min-Max Scaling and Z-score normalization for feature distributions. It would be helpful if the authors could justify the choice of these methods and explain when each one is applied. Providing examples of different scenarios where these methods are preferred would help readers better understand the reasoning behind their use and improve the transparency of the methodology.

2- There appears to be some confusion regarding the use of Min-Max Scaling and Feature Scaling in the paper. The authors should clarify the differences between these methods and explain why both were applied. If these are indeed distinct steps, a clear justification for their combined use is needed to avoid potential conflicts in the methodology. The readers might find it difficult to determine whether Min-Max Scaling or neural networks and gradient-based methods were used, so clarifying this point would enhance the clarity of the work.

3- In Algorithm 1, there is no clearly defined stop criterion, which could potentially lead to unnecessarily long running times if the iteration value (T) is set too high. For example, setting T to a large value may cause the algorithm to run for many iterations, even though convergence might occur much earlier. A predefined evaluation metric or maximum number of iterations should be established to ensure that the algorithm converges efficiently to an optimal solution.

4- In Section 3.3.1, the authors describe the CNN-LSTM architecture but do not specify important details such as the number of layers in both the CNN and LSTM networks, the number of neurons in each layer, or how these models are combined. Including such details is essential for ensuring that the methodology can be replicated. A diagram or figure that illustrates the architecture would be beneficial. Using tools like plot_model from tensorflow.keras.utils to visualize the model or providing a model summary would help improve transparency and enable other researchers to replicate the work.

5- The authors mention using transfer learning based on pre-trained weights, but do not specify which pre-trained model was used. It would be helpful to include this information so that readers can understand the specific model from which the weights were transferred. Additionally, further details on how transfer learning was implemented in the context of this work would add depth to the methodology section.

6- The paper does not describe how the proposed model was fine-tuned after the initial training. It would be beneficial for the authors to elaborate on the fine-tuning process, including any adjustments made to hyperparameters, learning rates, or other aspects of the model to improve its performance.

7- The authors state that the CNN-LSTM architecture was implemented in Python using either TensorFlow or PyTorch. It is important to be more precise and state explicitly which framework was used for the implementation. This clarification will help avoid any ambiguity and ensure that the methodology is reproducible.

8- The authors use two different datasets in their experiments: the CloudSim tool data and the Google Cluster Trace dataset. However, there is no mention of the performance of the CNN-LSTM model in terms of Mean Squared Error (MSE) or other relevant metrics. The authors should include results on MSE or similar performance measures to assess the model's predictive capability.

Reviewer #2: The paper aims to address the challenges of dynamic and efficient load balancing in cloud computing environments. Traditional methods often fall short in managing fluctuating workloads, leading to suboptimal resource utilization and increased operational costs. To tackle this, the authors propose SLADRO (Smart Load Adaptive Distribution with Reinforcement and Optimization), a novel framework integrating advanced machine learning and optimization techniques.

The paper presents a well-developed approach with strong results, but addressing the below comments will further strengthen the clarity and rigor of the presented work.

Problem Statement: The introduction needs to clearly state the gap in existing solutions and how SLADRO uniquely addresses this gap. While contributions are listed, their connection to specific limitations in current methods is not explicit.

Figures and Visualizations: Several figures (e.g., throughput, makespan) are mentioned but not adequately explained in the provided text. Ensure all figures are included and provide sufficient discussion on their relevance to the findings. The figure captions (e.g., Fig 4, Fig 5) should be more descriptive to effectively convey the role of each figure without relying solely on the text. Consider adding details about what specific scenario or comparison is illustrated in each figure.

Comparative Analysis: The manuscript claims SLADRO outperforms other methods, but the benchmarks (e.g., Round Robin, Least Connections) are relatively basic. Include comparisons with more advanced techniques like hybrid metaheuristics or AI-driven approaches (e.g., DPSO-GA, HBA-Z).

Scalability Testing: There is limited discussion on how SLADRO performs under varying workloads or in larger-scale cloud environments. Address scalability and robustness in more detail.

DRL Implementation: The description of the DQN-based scheduling approach is detailed but lacks clarity on hyperparameter settings and the exploration-exploitation balance (e.g., epsilon decay in the epsilon-greedy policy).

Feature Selection: While OOA-PSO is presented as a novel approach, the paper does not sufficiently justify why it is superior to other optimization techniques for feature selection.

The manuscript contains grammatical errors and awkward phrasing. For instance, "Fig 3 focuses on throughput..." and "Moving to 5, which shows the makespan..." lack professional tone and clarity. A thorough language edit is necessary.

Although energy consumption is highlighted as a key metric, there is insufficient explanation of how SLADRO achieves lower energy usage compared to baseline methods. Expand on the mechanisms that contribute to energy efficiency.

6. PLOS authors have the option to publish the peer review history of their article (what does this mean?). If published, this will include your full peer review and any attached files.

Reviewer #1: No

Reviewer #2: No

---

## [Decision Letter · Decision Letter 1]

5 Feb 2025

PONE-D-24-52330R1Smart Load Balancing in Cloud Computing: Integrating Feature Selection with Advanced Deep Learning ModelsPLOS ONE

Dear Dr. Fraihat,

Thank you for submitting your manuscript to PLOS ONE. After careful consideration, we feel that it has merit but does not fully meet PLOS ONE’s publication criteria as it currently stands. Therefore, we invite you to submit a revised version of the manuscript that addresses the points raised during the review process. When revising your paper, please ensure that you thoroughly address the referees' comments.

We look forward to receiving your revised manuscript.

Kind regards,

Davide La Torre

Academic Editor

PLOS ONE

Journal Requirements:

Reviewers' comments:

Reviewer's Responses to Questions

**Comments to the Author**

1. If the authors have adequately addressed your comments raised in a previous round of review and you feel that this manuscript is now acceptable for publication, you may indicate that here to bypass the “Comments to the Author” section, enter your conflict of interest statement in the “Confidential to Editor” section, and submit your "Accept" recommendation.

Reviewer #1: All comments have been addressed

Reviewer #2: (No Response)

2. Is the manuscript technically sound, and do the data support the conclusions?

Reviewer #1: Yes

Reviewer #2: Yes

3. Has the statistical analysis been performed appropriately and rigorously? 

Reviewer #1: Yes

Reviewer #2: No

4. Have the authors made all data underlying the findings in their manuscript fully available?

Reviewer #1: Yes

Reviewer #2: Yes

5. Is the manuscript presented in an intelligible fashion and written in standard English?

Reviewer #1: Yes

Reviewer #2: No

6. Review Comments to the Author

Reviewer #1: The authors have addressed my questions satisfactorily; therefore, I have no issues to accepting this paper for publication in the PLOS ONE journal.

Reviewer #2: While the authors have responded to the majority of my comments, I still have the following comments left unaddressed:

Comparative Analysis: More discussion is needed on the limitations of SLADRO compared to other state-of-the-art hybrid models, especially regarding computational cost and scalability.

SLADRO is now compared against DPSO-GA and HBA-Z, but the discussion remains limited. The authors should provide statistical significance testing (e.g., t-tests) to validate SLADRO’s superiority over DPSO-GA and HBA-Z methods.

Scalability Testing: The manuscript now mentions large-scale workloads, but no experiments are provided on how SLADRO performs under extreme workloads. The authors should include tests on larger-scale datasets (e.g., Azure or AWS workload traces).

Justification for OOA-PSO/Features Selection: The authors compare OOA-PSO with GA, PSO, and ACO, showing superior feature selection accuracy. The authors should explain why OOA-PSO works better for this specific problem beyond raw performance.

Energy Efficiency Explanation: The authors mention that SLADRO improves energy efficiency but do not explain the mechanism in detail. They should discuss how DRL and feature selection contribute to energy efficiency (e.g., fewer VM migrations, lower idle CPU usage).

Reproducibility: While the implementation details are robust, additional clarity on hyperparameter tuning, training duration, and computational resources used would improve reproducibility.

Writing & Formatting Issues: There are minor typographical errors and awkward phrasing in some sections (e.g., "Litarture review" should be "Literature Review").

I also recommend the following:

Justify the choice of ϵ and Tmax in Algorithm 1 (stop criterion).

Explain why transfer learning is necessary in this problem.

Add statistical significance testing for performance comparisons.

Provide additional scalability experiments for large cloud workloads.

Expand the discussion on energy efficiency mechanisms.

7. PLOS authors have the option to publish the peer review history of their article (what does this mean?). If published, this will include your full peer review and any attached files.

Reviewer #1: No

Reviewer #2: No

---

## [Decision Letter · Decision Letter 2]

22 Jul 2025

Smart Load Balancing in Cloud Computing: Integrating Feature Selection with Advanced Deep Learning Models

PONE-D-24-52330R2

Dear Dr. Fraihat,

We’re pleased to inform you that your manuscript has been judged scientifically suitable for publication and will be formally accepted for publication once it meets all outstanding technical requirements.

Kind regards,

Davide La Torre

Academic Editor

PLOS ONE

Additional Editor Comments (optional):

Reviewers' comments:

Reviewer's Responses to Questions

**Comments to the Author**

1. If the authors have adequately addressed your comments raised in a previous round of review and you feel that this manuscript is now acceptable for publication, you may indicate that here to bypass the “Comments to the Author” section, enter your conflict of interest statement in the “Confidential to Editor” section, and submit your "Accept" recommendation.

Reviewer #1: All comments have been addressed

Reviewer #2: All comments have been addressed

2. Is the manuscript technically sound, and do the data support the conclusions?

Reviewer #1: Yes

Reviewer #2: Yes

3. Has the statistical analysis been performed appropriately and rigorously? 

Reviewer #1: Yes

Reviewer #2: Yes

4. Have the authors made all data underlying the findings in their manuscript fully available?

Reviewer #1: Yes

Reviewer #2: Yes

5. Is the manuscript presented in an intelligible fashion and written in standard English?

Reviewer #1: Yes

Reviewer #2: Yes

6. Review Comments to the Author

Reviewer #1: (No Response)

Reviewer #2: Thank you for your submission. I appreciate the time and efforts you l have spent to revise the manuscript. I'm satisfied with the edits and responses to my comments.

7. PLOS authors have the option to publish the peer review history of their article (what does this mean?). If published, this will include your full peer review and any attached files.

Reviewer #1: No

Reviewer #2: **Yes: **Osama Al-Baik

---

## [Editor Report · Acceptance letter]

PONE-D-24-52330R2

PLOS ONE

Dear Dr. Fraihat,

I'm pleased to inform you that your manuscript has been deemed suitable for publication in PLOS ONE. Congratulations! Your manuscript is now being handed over to our production team.

Kind regards,

on behalf of

Dr. Davide La Torre

Academic Editor

PLOS ONE